

# Distributions and sources of gaseous and particulate low molecular weight monocarboxylic acids in a deciduous broadleaf forest from northern Japan

Tomoki Mochizuki[1,a], Kimitaka Kawamura[1,b], Yuzo Miyazaki[1], Suresh K.R. Boreddy[1]

[1]Institute of Low Temperature Science, Hokkaido University, Sapporo, Japan
[a]Now at School of Food and Nutritional Science, University of Shizuoka, Shizuoka, Japan
[b]Now at Chubu Institute for Advanced Studies, Chubu University, Kasugai, Japan

*Correspondence to*: K. Kawamura (kkawamura@isc.chubu.ac.jp)

**Abstract.** To better understand the distributions of low molecular weight (LMW) monocarboxylic acids (monoacids) and their sources in the forest, we conducted simultaneous collection of gaseous and particulate samples at a deciduous broadleaf forest site in northern Japan. LMW normal ($C_1$–$C_{10}$), branched chain ($iC_4$–$iC_6$), hydroxyl (lactic and glycolic) and aromatic (benzoic) monoacids were detected in the gas and particle phases. The dominant LMW monoacids in gas phase were formic (mean: 953 ng m$^{-3}$) and acetic (528 ng m$^{-3}$) acids. In particle phase, we found that isopentanoic (159 ng m$^{-3}$) and acetic (104 ng m$^{-3}$) acids are dominant species together with lactic acid. Concentrations of LMW monoacids did not correlate with $SO_4^{2-}$ that was used as an anthropogenic tracer, indicating that LMW monoacids are derived from the local sources within the forest ecosystem. Concentrations of $C_1$–$C_6$ monoacids in gas phase showed positive correlations ($r^2 = 0.21$–$0.91$) with isobutyric acid ($iC_4$), which is produced by soil microorganisms. These monoacids are closely linked to the microbial process in soils. Isopentanoic acid in particle phase showed a positive correlation with lactic acid ($r^2 = 0.98$), which is produced by soil microbes. The observed high abundances of isopentanoic acid are involved with soil microbial activity. We found that acetic acid in particle phase positively correlated with nonanoic acid ($C_9$) ($r^2 = 0.63$), suggesting that formation of acetic and nonanoic acids are associated with the oxidation of unsaturated fatty acids. We found that forest floor with soil microbes contributes to the emissions of gaseous and particulate LMW monoacids. Our results suggest that forest ecosystem is an important source of organic gases and aerosols in the atmosphere.

## 1 Introduction

Homologous series ($C_1$–$C_{10}$) of low molecular weight (LMW) monocarboxylic acids (monoacids) are known to exist in the atmosphere as gas and particle phases (e.g., Kawamura et al., 1985; 2000; Liu et al., 2012). They have been reported from urban (Kawamura et al., 2000), forest (Andreae et al., 1988), marine (Miyazaki et al., 2014; Boreddy et al., 2017), and Arctic samples (Legrand et al., 2004). Short chain monoacids such as formic and acetic acids are dominant chemical species in the atmosphere. LMW monoacids and their salts in aerosols are water-soluble and thus can act as cloud condensation nuclei



(CCN), contributing to the radiative forcing directly or indirectly (Kanakidou et al., 2005) and affecting the radiation budget of the earth's atmosphere. On the other hand, high abundances of LMW monoacids in the troposphere can adversely affect air quality and human health and also increase the acidity of rainwater (Keene et al., 1983; Kawamura et al., 1996).

LMW monoacids are directly emitted from fossil fuel combustion and biomass- and biofuel-burning (Kawamura et al., 1985; Paulot et al., 2011) and terrestrial vegetation (Kesselmeier et al., 1997; Jardine et al., 2011). In addition, secondary production from photochemical oxidations of biogenic volatile organic compounds (VOCs) such as isoprene and anthropogenic VOCs such as acetylene and ethane are important sources of LMW monoacids (Paulot et al., 2011). Recently, Stavrakou et al. (2012) conducted satellite measurement of formic acid on a global scale. They suggest that boreal and tropical forests are important sources of formic acid in the troposphere. In model experiment, Paulot et al. (2011) estimated that global source of formic and acetic acids are ~1200 Gmol year$^{-1}$ and ~1400 Gmol year$^{-1}$, respectively, however, these values are highly uncertain.

In our previous study on organic acids, normal ($C_1$-$C_{10}$), branched chain ($iC_4$-$iC_6$), and hydroxy (lactic and glycolic) monoacids were detected in gas, aerosol and snow pit samples (Kawamura et al., 2000; Mochizuki et al., 2016; 2017a). In particular, branched chain ($iC_5$) and hydroxy (lactic) monoacids were abundantly detected in aerosol samples from North East China (Mochizuki et al., 2017a). Detected branched chain ($iC_4$ and $iC_5$) and hydroxy (lactic) monoacids are derived from microorganisms and plant tissues. However, these monoacids have not been reported in the previous studies from the forest atmosphere, in which ion chromatograph was mainly used and the species detected are generally limited to formic and acetic acids. Because branched chain and hydroxy monoacids are highly water-soluble, they have a potential to change the hygroscopic properties of atmospheric particles. In addition, there is no study on gas/particle partitioning of normal ($C_1$–$C_{10}$), branched chain ($iC_4$–$iC_6$), and hydroxy (lactic and glycolic) monoacids in the forest atmosphere. Therefore, the study of LMW monoacids in forest floor is important.

In this study, we collected gas and particle samples in a deciduous broadleaf forest in northern Japan in summer. To better understand the distributions and sources of LMW monoacids, samples were analyzed for normal ($C_1$–$C_{10}$), branched ($iC_4$–$iC_6$), hydroxyl (lactic and glycolic), and aromatic (benzoic) monoacids in both gas and particle phases, along with inorganic ions in particle phase. We discuss the importance of monoacid-enriched aerosols and their possible sources in the forest atmosphere.

## 2 Experimental

The Sapporo forest meteorology research site (SAP) (42º59' N, 141º23' E, 182 m a.s.l.) is located in Sapporo, Hokkaido, Japan (147 ha) on a hilly area neighbouring urban district of Sapporo city (Figure 1). Residential area is located north, east and west of the site. The forest type is mature secondary deciduous broadleaf forest. The major tree is Japanese white birch (*Betula platyphylla* var. *japonica*) and a Japanese oak (*Quercus mongolica* var. *grosse serrata*). The major understory is a dwarf bamboo (*Sasa kurilensis* and *Sasa senanensis*). Ambient temperature, relative humidity, UV-A, wind speed, wind direction, and precipitation were measured on a meteorological tower (Figure 2). Details of the micrometeorological





measurements and site information have been described in Yamanoi et al. (2015) and Miyazaki et al. (2012a; 2012b). During the campaign period (June to July, 2010), ambient air temperature ranged from 18 °C to 26 °C (average: 21±2.3 °C), whereas relative humidity ranged from 69% to 96% (average: 87±7.9%). UV-A was high during the first half of the measurement period (except for 7 July) and low during the second half of the measurement period. The dominant wind direction throughout the sampling period was from east and south. Wind speed ranged from 0.2 to 0.6 m s$^{-1}$ (average: 0.4 m s$^{-1}$). Precipitation occurred in the morning of 1 July (11 mm), in the evening of 4 July (1.2 mm), and in the morning of 8 July (6.6 mm).

Samplings were conducted from 28 June to 8 July 2010. The samples were collected for 15 hours (5:00–20:00 LT) in daytime (n = 11) and 9 hours (20:00–5:00 LT) in nighttime (n = 11). Total suspended particle (TSP) and gaseous organic acids were collected using a low-volume air sampler equipped with the two-stage filter packs (URG-2000-30FG) at a flow rate of 15 L min$^{-1}$ (Kawamura et al., 1985). The particles were collected onto precombusted (450°C, 6 hours) quartz-fiber filters (47 mm diameter) (first stage), whereas gaseous organic acids were collected on the quartz-fiber filter impregnated with potassium hydroxide (KOH) (second stage). The KOH impregnated filters were prepared by rinsing the precombusted quartz filter in a 0.2 M KOH solution and then dried in an oven at 80 °C. Each filter was placed in a clean glass bottle with a Teflon-lined screw cap. After the sampling, the filter samples were stored in a freezer room at -20 °C prior to analysis. Semi-volatile organic acids collected on the first filter may evaporate, causing negative artifacts. On the other hand, second filter may absorb organic vapors evaporated from the first filter, causing positive artifacts. Although such artifacts are possible for any filter-based measurements for ambient conditions, these effects are limited (Kawamura et al., 1985).

LMW monoacids were determined as p-bromophenacyl esters using a capillary gas chromatograph equipped with a flame ionization detector (GC-FID) and GC-mass spectrometer (GC-MS) (Kawamura and Kaplan, 1984; Mochizuki et al., 2017a). Briefly, an aliquot of filter (4.3 cm$^2$) was extracted for water-soluble organic compounds with ultrapure water (resistivity of > 18.2 MΩ cm) under ultrasonication. To remove particles, the water extracts were filtered through quartz wool packed in a Pasteur pipette. The pH of filtrates was adjusted to 8.5–9.0 with 0.05 M KOH solution. The samples were concentrated down to 0.5 mL using a rotary evaporator under vacuum at 50 °C. The concentrations were passed through a cation exchange resin (DOWEX 50W-X4, 100-200 mesh, K$^+$ form) packed in a Pasteur pipette. Free monocarboxylic acids were converted to organic acid salts (RCOO$^-$K$^+$). After confirming the pH of 8.5–9.0, the samples were dried using a rotary evaporator under vacuum at 50 °C, followed by blowdown with pure nitrogen gas. Organic acids were derivatized to p-bromophenacyl esters in acetonitrile (4 ml) with α,p-dibromoacetophenone (0.1 M, 50 µl) as a derivatization reagent and dicyclohexyl-18-crown-6 (0.01 M, 50 µl) as a catalyst at 80°C for 2 hours (Kawamura and Kaplan, 1984). In addition, OH functional groups in p-bromophenacyl esters of hydroxymonoacids were reacted with N,O-bis-(trimethylsilyl)trifluoroacetamide (BSTFA) with 1% trimethylsilyl chloride and 10 µl of pyridine at 70°C for 3 hours to derive trimethylsilyl (TMS) ethers of p-bromophenacyl esters (Kawamura et al., 2012).

p-Bromophenacyl esters and their TMS ethers were identified and quantified using a capillary gas chromatograph (HP GC6890, Hewlett-Packard, USA) equipped with a flame ionization detector and GC-mass spectrometer (Agilent GC7890A





and 5975C MSD, Agilent, USA). Details of the methods have been described in Kawamura and Kaplan (1984) and Kawamura et al. (2012). Recoveries of authentic monoacids ($C_1$–$C_{10}$, $iC_4$–$iC_6$, lactic, glycolic, and benzoic acids) spiked to a quartz filter were better than 80%. Analytical errors using authentic monoacids were within 12%.

To measure inorganic ions, a portion of quartz-fiber filter (first filter) was extracted with ultrapure water under ultrasonication. The extracts were passed through a membrane disk filter (0.22 μm, Millipore Millex-GV, Merck, USA). The filtrates were injected into an ion chromatograph (Model 761 compact IC, Metrohm, Switzerland) (Boreddy and Kawamura, 2015). We measured cations ($Na^+$, $NH_4^+$, $K^+$, $Mg^{2+}$, and $Ca^{2+}$) and anions ($F^-$, $MSA^-$, $Cl^-$, $NO_2^-$, $NO_3^-$, $PO_4^-$, and $SO_4^{2-}$) in aerosol samples. Inorganic ions were not detected in the field blanks.

## 3 Results

### 3.1 LMW monocarboxylic acids

Low molecular weight normal ($C_1$–$C_{10}$), branched chain ($iC_4$–$iC_6$), hydroxyl (lactic and glycolic), and aromatic (benzoic) monoacids were detected in gas and aerosol samples from a deciduous broadleaf forest in northern Japan (Table 1). Figure 3 shows mean concentrations of monoacids in gas and particle phases. In gas phase, formic acid was the dominant species (188–2266 ng m$^{-3}$, mean: 953 ng m$^{-3}$), followed by acetic acid (277–1595 ng m$^{-3}$, mean: 528 ng m$^{-3}$). In particulate phase, isopentanoic acid was found as the dominant species (36–1478 ng m$^{-3}$, mean: 159 ng m$^{-3}$), followed by acetic acid (22–263 ng m$^{-3}$, mean: 104 ng m$^{-3}$) and formic acid (2.2–216 ng m$^{-3}$, mean: 71 ng m$^{-3}$). Lactic acid, which is a hydroxyl monoacid, was generally the fourth most abundant LMW monoacid in particle samples (8.4–522 ng m$^{-3}$, median: 65 ng m$^{-3}$). Nonanoic acid ($C_9$) is the dominant species (1.5–25 ng m$^{-3}$, mean: 11 ng m$^{-3}$) in particle phase, except for formic and acetic acids (Figure 3).

Figure 4 shows day-night variations of selected monoacids in gas and particle phases. Gaseous formic acid did not show any day/night trend, whereas particulate formic acid showed a diurnal distribution with higher concentrations in nighttime than daytime. Abundances of gaseous acetic acid in daytime were higher than those in nighttime, whereas opposite trend was found for particle phase acetic acid, that is, particulate acetic acid was more abundant in nighttime than in daytime. Although, large day-to-day variations in formic and acetic acids were observed in both gas and particle phases, these acids did not show any temporal trend.

The highest concentrations of isopentanoic and lactic acids in particle phase were observed at night on 30 August. Temporal variation of isopentanoic acid in particle phase was similar to that of lactic acid. Gaseous and particulate concentrations of isopentanoic and lactic acids did not show any clear diurnal trend.

The particle-phase fractions ($F_p$) of individual monoacids were calculated as $F_p$ = P/(G+P), where P is particle-phase concentration and G is gas phase concentration. Table 1 summarizes mean $F_p$ of individual monoacids in daytime and nighttime in the deciduous broadleaf forest. $F_p$ of individual monoacids ranged from 0.04 ($C_3$) to 0.63 ($iC_5$) in daytime and 0.05 ($C_3$) to 0.69 ($iC_5$) in nighttime. Short-chain monoacids such as formic and acetic acids are largely present in gas phase,





except for isopentanoic acid. Nonanoic ($C_9$) and decanoic ($C_{10}$) acids are present not only in gas phase but also in particle phase. Lactic acid is largely present in aerosol phase in the forest atmosphere.

## 3.2 Inorganic ions in particles

We detected cations ($Na^+$, $NH_4^+$, $K^+$, $Mg^{2+}$, and $Ca^{2+}$) and anions ($NO_3^-$, $SO_4^{2-}$, $MSA^-$, $Cl^-$, $NO_2^-$, and $F^-$) in particle samples

from a deciduous broadleaf forest. $SO_4^{2-}$ (mean: 2320 ng m$^{-3}$) is major anion and $NH_4^+$ (mean: 972 ng m$^{-3}$) is major cation. Concentrations of major inorganic ions did not show clear diurnal and temporal variations. pH of the water extracts from particle samples ranged from 3.5 to 6.3 (mean: 5.0). The particle samples were always acidic.

## 4. Discussion

### 4.1 Possible sources of LMW monoacids

LMW monoacids have a variety of anthropogenic and biogenic sources. This forest site is located a few kilometers south of Sapporo city. The dominant wind direction was from east and south throughout the sampling period, and wind speed was very low (average: 0.4 m s$^{-1}$). In addition, we compared the concentrations of individual monoacids together with $SO_4^{2-}$: an anthropogenic aerosol tracer to evaluate the influence of anthropogenic air mass transport from urban area. We confirmed that individual monoacids in both gas and particle phases did not show a correlation with $SO_4^{2-}$ ($r^2 < 0.21$). These results

indicate that the majority of gases and aerosols of LMW monoacids were derived from the local sources within the forest ecosystem.

     In gas phase, isobutyric acid ($iC_4$) showed positive correlations with $C_1$ (day: $r^2 = 0.36$, night: $r^2 = 0.21$), $C_2$ (day: $r^2 = 0.53$, night: $r^2 = 0.43$), $C_3$ (day: $r^2 = 0.76$, night: $r^2 = 0.64$), $C_4$ (day: $r^2 = 0.82$, night: $r^2 = 0.80$), $C_5$ (day: $r^2 = 0.91$, night: $r^2 = 0.81$) and $C_6$ (day: $r^2 = 0.72$, night: $r^2 = 0.74$) monoacids (Figure 5). Branched chain monoacids can be produced by

microbiological (e.g., bacteria and actinomyces) processes (Allison, 1978; Hafner et al., 1990). These microorganisms live in soil. On the other hand, lactic acid ($hC_3$) in daytime did not show positive correlations with $C_1$–$C_6$ monoacids ($r^2 < 0.004$), whereas lactic acid in nighttime shows a positive correlation with $C_3$–$C_6$ ($r^2 = 0.45$–0.65) although they were rather scattered. Bacteria (*lactobacillus*) are known to produce lactic acid (Cabredo et al., 2009), which mainly exist in soil (Huysman and Verstraete, 1993). Variety of emission sources of $C_1$–$C_6$ monoacids may exist in forest soil. We suggest that gaseous $C_1$–$C_6$

monoacids are directly emitted from the forest floor where soil microorganisms contribute to the emissions of gaseous $C_1$–$C_6$ monoacids.

     Plant leaves have several synthetic pathways of formic and acetic acids (e.g., Christensen and MacKenzie, 2006; Liedvogel and Stumpf, 1982). Gaseous formic and acetic acids are directly emitted from plant leaves (Kesselmeier and Staudt, 1999). These emissions depend on temperature and solar radiation (Kesselmeier et al., 1997). In this study,

concentrations of formic and acetic acids in gas phase showed a weak positive correlation or no correlation in daytime with



ambient temperature ($C_1$: $r^2 = 0.14$, $C_2$: $r^2 = 0.28$) and UV-A ($C_1$: $r^2 = 0.0002$, $C_2$: $r^2 = 0.20$). The contribution of soil emissions may be much greater than plant emissions in a deciduous broadleaf forest.

In particle phase, a positive correlation was observed between acetic acid and nonanoic acid (day: $r^2 = 0.63$, night: $r^2 = 0.63$) (Figure 6). Unsaturated fatty acids (UFAs) such as oleic ($FA_{18:1}$) and linoleic ($FA_{18:2}$) acids are generally present in terrestrial higher plants and soil fungi (Yokouchi and Ambe, 1986; Kaur et al., 2005). Nonanoic ($C_9$) and hexanoic ($C_6$) acids are produced by the heterogenous oxidation of $FA_{18:1}$ and $FA_{18:2}$ in aerosols, respectively, via the cleavage of a double bond at $C_9$-position (Yokouchi and Ambe, 1986; Kawamura and Gagosian, 1987). Longer-chain monoacids may produce acetic acid via photochemical breakdown with OH radicals. UFAs may contribute to the formation of acetic acid in aerosol in a deciduous broadleaf forest.

Particulate formic and acetic acids in daytime negatively correlated with pH of the water extracts from aerosol samples ($C_1$: $r^2 = 0.35$, $C_2$: $r^2 = 0.43$) (Figure 7). Acidity of aerosol particles can enhance the formation of SOA via organic precursors and subsequent heterogeneous reactions in aerosol liquid phase (e.g., Jang et al., 2002; Pathak et al., 2011). Secondary formation of formic and acetic acids is associated with photo-oxidation of biogenic volatile organic compounds (BVOCs) such as isoprene and monoterpene (Ervens et al., 2008; Kawamura et al., unpublished data). Based on the field measurement, aerosol acidity is important for the formation of organic aerosols such as isoprene SOA tracers, α-pinene SOA tracers and dicarboxylic acids (oxalic acid) from BVOCs in the forest atmosphere (Mochizuki et al., 2015; 2017b). Formic and acetic acids are intermediate products in the complex heterogeneous oxidation of BVOCs. Lower pH may promote the secondary formation of formic and acetic acids from BVOCs in the forest atmosphere (Kawamura et al., unpublished data).

Formic and acetic acids in particle phase show clear day-night variations and their concentrations in particle phase in nighttime were higher than those in daytime. Because wind speed in nighttime (average: 0.3 m s$^{-1}$) was comparable to that in daytime (average: 0.4 m s$^{-1}$) and anti-correlation was not observed between formic and acetic acids and wind speed ($r^2 <$ 0.01), accumulation of formic and acetic acids in the forest canopy followed by the increased concentrations may not show any clear day-night trend. Formic and acetic acids are later-generation products of the oxidation of BVOCs (e.g., Ervens et al., 2008). BVOCs or their earlier-generation products may be continuously oxidized at night in the forest atmosphere.

Isopentanoic acid can be produced by bacteria (Allison, 1978). A positive correlation was observed between lactic acid and isopentanoic acid in particle phase ($r^2 = 0.98$). Isopentanoic and lactic acid did not show a correlation with other LMW monoacids detected in particle phase ($r^2 < 0.17$). Hence, we suggest that lactic and isopentanoic acids are linked in the biosynthetic processes and formic and acetic acids in particle phase are mainly derived from secondary formations via heterogeneous oxidation on aerosol surfaces.

## 4.2 Gas/particle partitioning of LMW monoacids

Generally, $F_p$ increases with an increase in carbon numbers of monoacid (Yatavelli et al., 2014) due to lower vapor pressures of higher MW organic acids. The smaller $F_p$ of formic (daytime: 0.08) and acetic (daytime: 0.14) acids can be explained by higher vapor pressure ($C_1$: $5.6 \times 10^{-2}$ atm, $C_2$: $2.1 \times 10^{-2}$ atm) among LMW monoacids determined. The values of $F_p$ for formic



and acetic acids measured in a deciduous broadleaf forest in daytime are comparable to those reported from the Pacific Ocean ($C_1$: $F_p$ = 0.04, $C_2$: $F_p$ = 0.06) (Miyazaki et al., 2014) and urban Los Angeles ($C_1$: $F_p$ = 0.16, $C_2$: $F_p$ = 0.06) (Kawamura et al., 2000), in which the same sampling and analytical protocols were used.

The larger $F_p$ of lactic acid (daytime: 0.60, nighttime: 0.69) was observed. This level is comparable to that reported for the Pacific Ocean ($F_p$ = 0.82) (Miyazaki et al., 2014). Even though the vapor pressure of lactic ($5.3 \times 10^{-4}$ atm) and isopentanoic ($5.8 \times 10^{-4}$ atm) acids are higher than that of $C_5$–$C_{10}$ monoacids (vapor pressure: $1.6 \times 10^{-4}$–$1.6 \times 10^{-10}$ atm), the $F_p$ of lactic and isopentanoic acids were larger than that of $C_5$–$C_{10}$ monoacids ($F_p$: 0.11–0.50). These results suggest that particulate lactic and isopentanoic acids may be derived from primary sources in forest ecosystem via microbial activity.

We investigated the effects of ambient temperature on gas/particle partitioning of LMW monoacids. $F_p$ of formic and acetic acids were found to decrease with ambient temperature ($C_1$: $r^2$ = 0.49, $C_2$: $r^2$ = 0.60) (Figure 8), whereas other LMW monoacids did not show a correlation with ambient temperature ($r^2$ < 0.37), except for butyric acid ($C_4$) in daytime ($r^2$ = 0.70). Khan et al. (1995) reported that ambient temperature is an important factor to determine the gas/particle partitioning of organic acids. Lower temperature promotes the transfer of gas phase formic and acetic acids to aerosol phase in daytime, which is consistent with Henry's Law constants.

Liquid water contents (LWC) of aerosols were estimated using ISORROPIA-II model (Fountoukis and Nenes, 2007) with input compositions of inorganic ions and meteorological data. We found that $F_p$ of LMW monoacids did not show good correlations with LWC ($r^2$ < 0.24), suggesting LWC may not be an important factor to control the gas/particle partitioning of LMW monoacids in the forest atmosphere in this study.

Gaseous organic acids react with alkaline particles such as calcium and consequently the reactions enhance the
partitioning of gaseous organic acids to aerosol phase (Alexander et al., 2015). We calculated total cation equivalents ($Na^+$, $NH_4^+$, $K^+$, $Mg^{2+}$, and $Ca^{2+}$) minus total anion equivalents ($F^-$, $MSA^-$, $Cl^-$, $NO_2^-$, $NO_3^-$, and $SO_4^{2-}$) including monoacids detected, although $CO_3^-$, $HCO_3^-$, and unidentified organic anions were not considered. Total cations were higher than total anions. Excess cations exist in the aerosol. No positive correlation were observed ($r_2$ < 0.04) between $F_p$ of individual LMW monoacids and excess cations. This result indicates that excess cations are not an important factor to control the gas/particle
partitioning of LMW monoacids in the forest atmosphere. Our results suggest that ambient temperature is an important factor to control the gas/particle partitioning of formic and acetic acids in the forest atmosphere.

### 4.3 Comparison with other studies

The gas phase mean concentration levels of formic and acetic acids ($C_1$: 953 ng m$^{-3}$, $C_2$: 528 ng m$^{-3}$) in a deciduous broadleaf forest from northern Japan are lower than those reported from tropical forest from Amazon ($C_1$: 3400ng m$^{-3}$, $C_2$: 5900 ng m$^{-3}$) (Andreae et al., 1988) and urban air in Pasadena ($C_1$: 4700 ng m$^{-3}$) (Yuan et al., 2015) and in Los Angeles ($C_2$: 1800 ng m$^{-3}$) (Kawamura et al., 2000), but higher than those reported from the Pacific Ocean ($C_1$: 55 ng m$^{-3}$, $C_2$: 122 ng m$^{-3}$) (Miyazaki et al., 2014) and Antarctica ($C_1$: 75 ng m$^{-3}$, $C_2$: 92 ng m$^{-3}$) (Legrand et al., 2012). Particle phase concentrations of formic (71 ng m$^{-3}$) and acetic (104 ng m$^{-3}$) acids from a deciduous broadleaf forest in northern Japan are lower than those from urban



Los Angeles ($C_1$: 163 ng m$^{-3}$, $C_2$: 120 ng m$^{-3}$) (Kawamura et al., 2000), Beijing ($C_1$: 370 ng m$^{-3}$, $C_2$: 350 ng m$^{-3}$) (Wang et al., 2005), but higher than those reported in tropical forests from Amazon ($C_1$: 46 ng m$^{-3}$, $C_2$: 48 ng m$^{-3}$) (Andreae et al., 1988) and from Ocean in the Pacific Ocean ($C_1$: 2 ng m$^{-3}$, $C_2$: 8 ng m$^{-3}$) (Miyazaki et al., 2014) and in the southern Bay of Bengal ($C_1$: 21 ng m$^{-3}$, $C_2$: 25 ng m$^{-3}$) (Boreddy et al., 2017). These comparisons demonstrate that ecosystem of deciduous broadleaf forest is a source of atmospheric formic and acetic acids in both gas and aerosol phases.

Interestingly, the level of particle phase isopentanoic acid (159 ng m$^{-3}$) is higher than that of formic (71 ng m$^{-3}$) and acetic (104 ng m$^{-3}$) acids from a deciduous broadleaf forest in northern Japan during the campaign period. Average concentration of isopentanoic acid is higher than that in marine aerosol samples from the Bay of Bengal (12 ng m$^{-3}$) (Boreddy et al., 2017) and isopentanoic acid has not been reported from urban Los Angeles (Kawamura et al., 2000). On the other hand, high abundances of isopentanoic acid in particle phase from Mt. Tai (China) were observed during field burning of agricultural wastes (331 ng m$^{-3}$) (Mochizuki et al., 2017). Lactic acid in particle phase (65 ng m$^{-3}$) was the fourth most abundant monoacid in a deciduous broadleaf forest from northern Japan. Average particle phase concentration of lactic acid in the forest atmosphere from northern Japan is higher than that in marine aerosol samples from the Pacific (33 ng m$^{-3}$) (Miyazaki et al., 2014), pasture samples (22 ng m$^{-3}$) and forest samples (9.2 ng m$^{-3}$) in Brazil (Graham et al., 2002). On the other hand, high abundances of lactic acid in particle phase from Mt. Tai (China) were observed during field burning of agricultural wastes (917 ng m$^{-3}$) (Mochizuki et al., 2017). Although formic and acetic acids are in general predominant organic species in the atmosphere, our study indicate that branched chain and hydroxy monoacids are also important organic acids in the deciduous broadleaf forest. Because reports on branched chain and hydroxy monoacids are limited, more study is needed to clarify the sources and emissions of these organic acids in the forest atmosphere.

## 4. Summary and Conclusions

We conducted simultaneous sampling of gaseous (G) and particulate (P) LMW monoacids in a deciduous broadleaf forest from northern Japan followed by p-bromophenacyl ester derivatization and gas chromatographic determination. LMW normal ($C_1$–$C_{10}$), branched chain (i$C_4$–i$C_6$), hydroxyl (lactic and glycolic), and aromatic (benzoic) monoacids were detected in both gas and aerosol samples studied. Formic and acetic acids were found as the dominant species in gas phase. Isopentanoic and acetic acids were detected as the dominant species in particle phase. Particle-phase fractions ($F_p$ = P/(P+G)) of formic (average: 0.10) and acetic (0.19) acids are low. Major LMW monoacids were present in gas phase, except for isopentanoic ($F_p$ = 0.68) and lactic acids ($F_p$ = 0.66). Concentrations of LMW monoacids did not correlate with $SO_4^{2-}$ (anthropogenic tracer), suggesting that forest ecosystem is an important source of LMW monoacids. Concentrations of $C_1$–$C_6$ monoacids in gas phase showed positive correlations with isobutyric acid (i$C_4$). Branched chain monoacids can be used as an indicator of soil microorganism processes. Gaseous $C_1$–$C_6$ monoacids are directly emitted from the forest floor associated with microorganism activities. Isopentanoic acid in particle phase showed positive correlation with lactic acid ($r^2$ = 0.98), which is produced by soil microbes. The observed high concentrations of isopentanoic acid are involved with soil microbial activity. Concentrations of acetic acid in particle phase showed a positive correlation with nonanoic acid ($C_9$), suggesting





that formation of $C_9$ acid is associated with the oxidation of unsaturated fatty acids. Our study demonstrated that deciduous broadleaf forest in northern Japan is an important source of LMW monoacids in gas and particle phases in the atmosphere. Forest floor including soil microorganisms contributes to the emissions of gaseous and particulate LMW monoacids. Our results may be useful to improve the understanding of organic aerosol formation in the forest.

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



**Figure captions**

Figure 1. Location of a forest site for air sampling.

Figure 2. Diurnal and temporal variations of (a) temperature and relative humidity, (b) UV-A, and (c) wind speed and wind direction in a deciduous broadleaf forest.

Figure 3. Average concentrations of LMW monocarboxylic acids in gas and particle phases.

Figure 4. Diurnal variations in the concentrations of major monocarboxylic acids in gas and particle phases and inorganic ions ($SO_4^{2-}$ and $NH_4^+$). Open circles indicate gas phase samples and solid diamonds indicate particle phase samples (Day: D, Night: N).

Figure 5. Concentrations of $C_1$–$C_6$ monoacids against isobutyric acid ($iC_4$) in gas phase.

Figure 6. Concentrations of formic and acetic acids in gas phase as a function of nonanoic acid.

Figure 7. Scatter plots of concentrations of formic and acetic acids in particle phase against pH of the water extracts from aerosol samples.

Figure 8. Particle-phase fractions ($F_p$) of formic and acetic acids against temperature.







**Figure 1. Location of a forest site for air sampling.**

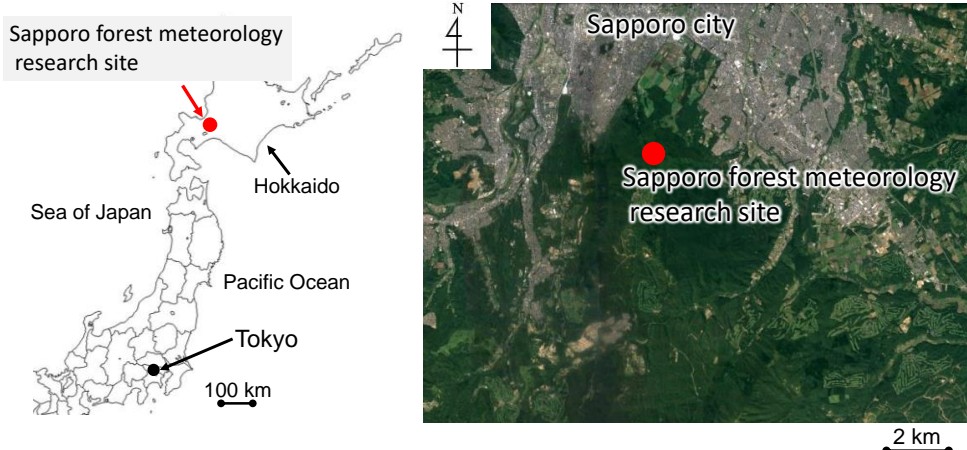

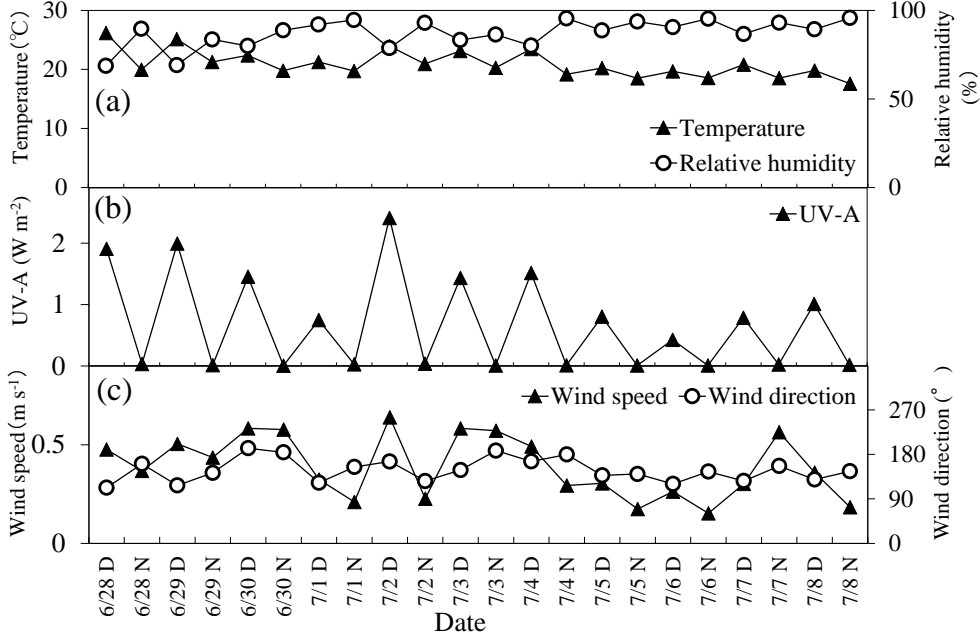

**Figure 2. Diurnal and temporal variations of (a) temperature and relative humidity, (b) UV-A, and (c) wind speed and wind direction in a deciduous broadleaf forest.**





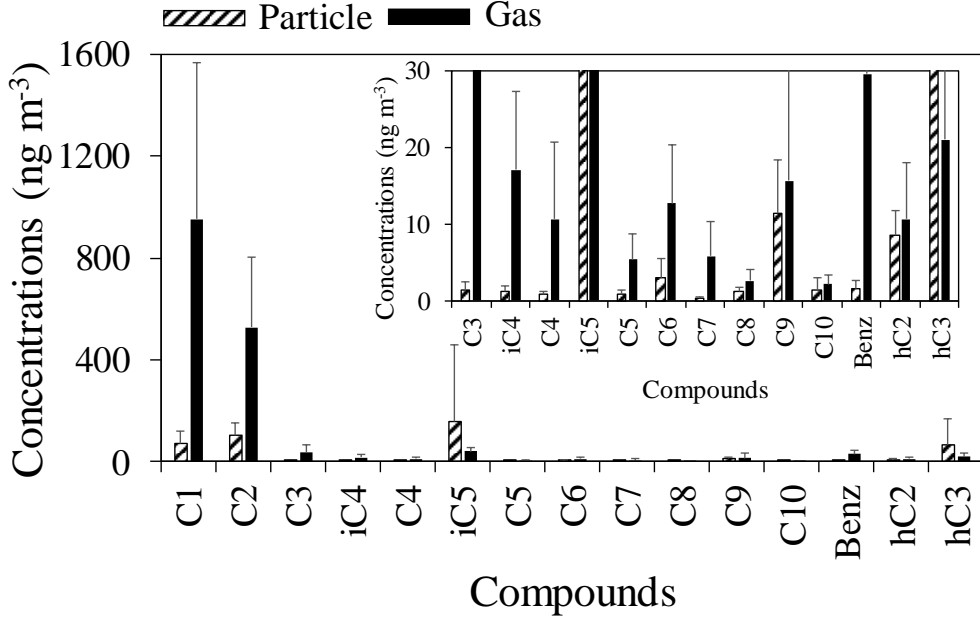

**Figure 3. Average concentrations of LMW monocarboxylic acids in gas and particle phases.**



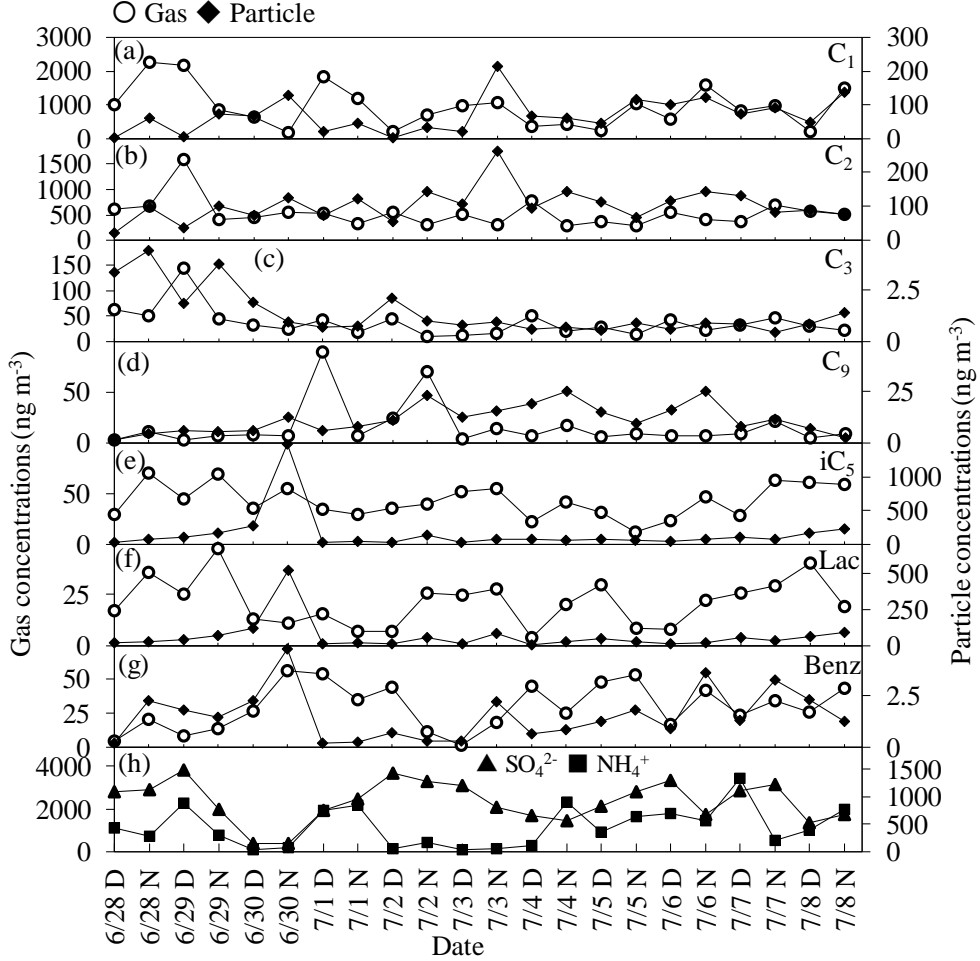

**Figure 4.** Diurnal variations in the concentrations of major monocarboxylic acids in gas and particle phases and inorganic ions ($SO_4^{2-}$ and $NH_4^+$). Open circles indicate gas phase samples and solid diamonds indicate particle phase samples (Day: D, Night: N).





Figure 5. Concentrations of $C_1$–$C_6$ monoacids against isobutyric acid (i$C_4$) in gas phase.




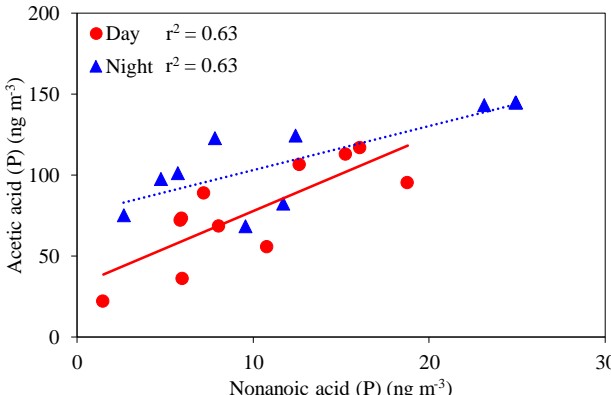

**Figure 6. Concentrations of formic and acetic acids in gas phase as a function of nonanoic acid.**

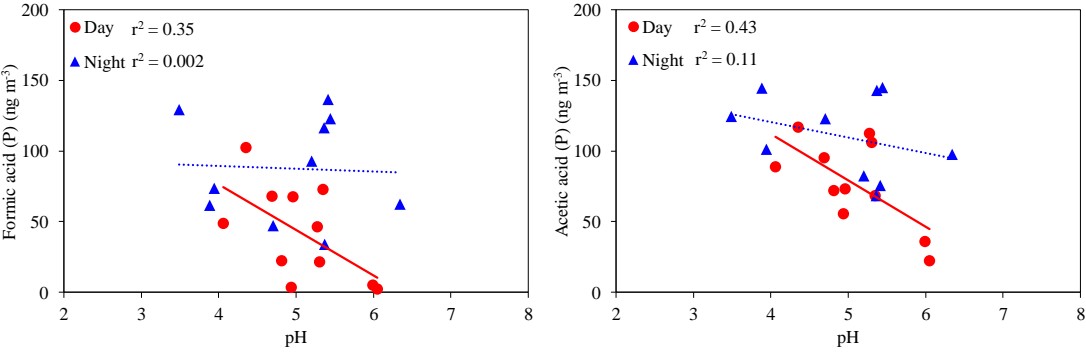

**Figure 7. Scatter plots of concentrations of formic and acetic acids in particle phase against pH of the water extracts from aerosol samples.**

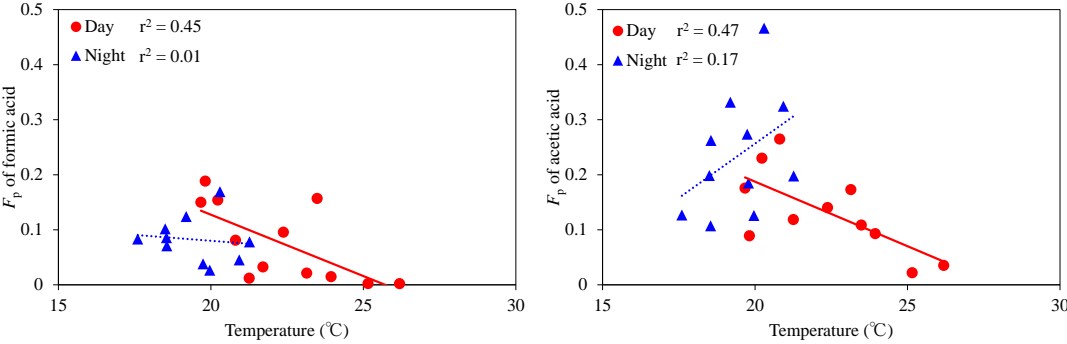

**Figure 8. Particle-phase fractions ($F_p$) of formic and acetic acids against temperature.**



**Table 1. Average concentrations (ng m⁻³) with standard deviation (S.D.) of LMW monocarboxylic acids in gas and particle phases and particle-phase fraction ($F_p$) in a deciduous broadleaf forest from northern Japan.**

| | Particle phase | | | | Gas phase | | | | $F_p$ | | | |
| | Day | | Night | | Day | | Night | | Day | | Night | |
| Organic acids | Ave. | STD | Ave. | STD | Ave. | STD | Ave. | STD | Ave. | STD | Ave. | STD |
|---|---|---|---|---|---|---|---|---|---|---|---|---|
| **Aliphatic acids** | | | | | | | | | | | | |
| Formic, $C_1$ | 41 | 32 | 99 | 52 | 832 | 627 | 1075 | 572 | 0.08 | 0.07 | 0.11 | 0.11 |
| Acetic, $C_2$ | 81 | 34 | 124 | 54 | 603 | 334 | 431 | 154 | 0.14 | 0.07 | 0.24 | 0.11 |
| Propionic, $C_3$ | 1.5 | 1.1 | 1.5 | 1.3 | 46 | 33 | 27 | 14 | 0.04 | 0.03 | 0.05 | 0.02 |
| Isobutyric, $iC_4$ | 1.1 | 0.8 | 1.3 | 0.7 | 18 | 11 | 16 | 8.4 | 0.07 | 0.05 | 0.08 | 0.04 |
| Butyric, $C_4$ | 0.8 | 0.4 | 0.9 | 0.4 | 13 | 12 | 7.5 | 5.1 | 0.07 | 0.04 | 0.13 | 0.09 |
| Isopentanoic, $iC_5$ | 86 | 72 | 226 | 419 | 41 | 20 | 49 | 17 | 0.63 | 0.19 | 0.69 | 0.14 |
| Pentanoic, $C_5$ | 0.7 | 0.3 | 1.2 | 0.3 | 5.6 | 3.7 | 5.0 | 3.0 | 0.14 | 0.11 | 0.23 | 0.10 |
| Hexanoic, $C_6$ | 1.6 | 1.0 | 4.6 | 2.7 | 12 | 6.3 | 13 | 8.8 | 0.13 | 0.08 | 0.28 | 0.10 |
| Heptanoic, $C_7$ | 0.2 | 0.2 | 0.3 | 0.2 | 4.5 | 3.0 | 7.0 | 5.4 | 0.11 | 0.06 | 0.13 | 0.13 |
| Octanoic, $C_8$ | 0.8 | 0.4 | 1.5 | 0.6 | 2.2 | 1.2 | 3.1 | 1.4 | 0.30 | 0.12 | 0.34 | 0.11 |
| Nonanoic, $C_9$ | 9.2 | 5.5 | 13 | 8.1 | 14 | 24 | 16 | 18 | 0.50 | 0.22 | 0.47 | 0.17 |
| Decanoic, $C_{10}$ | 1.6 | 1.2 | 1.3 | 1.9 | 1.7 | 0.9 | 2.6 | 1.2 | 0.49 | 0.21 | 0.34 | 0.23 |
| Sub total | 225 | | 475 | | 1594 | | 1652 | | | | | |
| **Hydroxyacids** | | | | | | | | | | | | |
| Glycolic, Glyco | 8.4 | 3.4 | 8.6 | 3.1 | 11 | 9.4 | 10 | 4.1 | 0.48 | 0.14 | 0.47 | 0.13 |
| Lactic, Lac | 38 | 33 | 90 | 145 | 24 | 21 | 23 | 12 | 0.60 | 0.17 | 0.68 | 0.16 |
| Sub total | 46 | | 99 | | 36 | | 33 | | | | | |
| **Aromatic acid** | | | | | | | | | | | | |
| Benzoic, Benz | 1.0 | 0.8 | 2.0 | 1.4 | 26 | 18 | 32 | 15 | 0.06 | 0.06 | 0.06 | 0.04 |