# Peer review of "Distributions and sources of low molecular weight monocarboxylic acids in gas and particle from a deciduous broadleaf forest in northern Japan"

_Atmospheric Chemistry and Physics, 2018_

## Referee Comment (RC1) · Anonymous Referee #1 · 11 Jun 2018

Manuscript describes the results from the measurements of low molecular weight acids in theforest air in Japan. These acids have effects on atmospheric chemistry, acidity and clouds, but their sources and concentrations in the air are poorly known. Even with very limited data set, study gives new knowledge on these poorly known compounds. However, major improvements on interpreting the results is needed.

Major comments: It would be very interesting to see how big fraction of the particulate mass these acids comprise. Could you add some information on particle mass?

[Figure]

More discussion on other possible sources are needed. For example production in the air in gas phase may be very important and also transport from anthropogenic or other sources.

Please discuss also on lifetime of these compounds and background levels. For example for acetic acid lifetime in the air is around two week while for decanoic acid only few hours.

Page 5, line 24-25: From only ambient air concentrations you cannot state that some are emitted from forest floor and soil microorganisms. Or do you have data on microorganisms? If so, please show it in manuscript. There are several different sources and sinks possibly affecting the concentration levels and for some compounds even background levels may be relatively high.

Page 5 and 6, lines 30and 1-2: Even ambient air concentrations of monoterpenes, with clear temperature and light dependent emissions, do not most often show direct correlation with temperature since there are several processes affecting on ambient concentration levels in the air (e.g. mixing layer height, vertical wind, background levels and reactivity). Therefore I recommend you to remove this statement on contribution of soil being higher than forest canopy.

Page 6, lines 10-18: This paragraph is very unclear. It was not clear to me, when you discuss on gas phase and when particulate. I would expect that lots of formic and acetic acids are formed in the gas phase reactions of BVOCs and then these formed acids partition into particulate phase. Of course some additional could then be formed also in particles or their surfaces.

Page 6, line 23-24: BVOCs with double bonds do react in the air with O3 and NO3 also during the night. But after first reactions they have lost the double bonds and continue reactions with OH radicals, which are formed only during the day. Double bonds from BVOCs are lost well-before they form formic and acetic acid and therefore I expect that formic and acetic acids are formed in the reactions of other VOCs only during the day. I

would expect that due to partitioning between gas and aerosol phase more is detected in particulate phase during colder the nights and then during the day when temperature increases they are evaporated. You should add discussion also on this.

Page 7, line 4-8: If lactic acid is more polar and more water soluble it will partition more to the particulate phase. Discuss also on this possibility.

Page 7, line 13. Effect of temperature applies also during nighttime.

More discussion on also sinks and lifetimes and effect of local chemistry is needed.

Page 8, line 29. Add references on branched chain monoacids being an indicator of soil microorganism processed.

Page 8, line 30: Based on the data, it cannot be stated that microorganisms are source of these acids. They can as well be produced in the local reactions of other emitted compounds

Page 8, line 32: Do you have any data on microbial activity? If not, remove this statement.

Page 9, line 3: Based on the data shown here, this cannot be stated. Maybe they give some indication that soil could be a source, but it cannot be stated.

Even with low concentrations C4-C10 acids can be very important and have higher local effect and higher emissions/sources that you would think by their concentrations. This is because their reactivity and lifetimes in the air are much shorter. Especially for these acids data on ambient concentrations and sources are very limited. Please, add more discussion also on these acids.

Could you add a table on the comparison with other studies?

Minor comments:

Language should be checked by some native speaker

**[ACPD](javascript:void(0))**
* * *
Interactive
comment

Page 4 lines 15-18. Frist you tell that isopentanoic acid is most dominant in particles and then nonanoic acid. Please, correct these sentences.

Page 4 line 25. What temporal trend? Please explain more clearly.

Fig. 4. It is very hard to deviate between the days. Could you add for example some dashed lines between the days?
* * *

---

## Referee Comment (RC2) · Anonymous Referee #2 · 19 Jul 2018

The authors of this paper report the concentrations of low molecular weight monocarboxylic acids in the gas and particle phase samples collected at a deciduous broad leaf forest site near Sapporo, northern Japan. They claim that acetic acid, isopentanoic acid, and lactic acid are the major particle phase monoacids at the site and they claim that isopentanoic acid and lactic acid originate from soil bacteria activities because they correlate well with gas phase isobutyric acid that may be produced by soil bacteria. While some of the data presented here may be interesting for people working in this field, the paper has major flaws as there are too many speculative statements

that do not stand up to scrutiny. I recommend major revision before this paper could be published in ACP.

Page 5 Line 12: It is not clear to me why the authors chose sulfate ion as an anthropogenic tracer for local urban emissions. Is the total sulfate ion or is it nss-sulfate? How does the authors exclude the influence of long range transport from China or Russia? This should be clarified.

Page 5 Lines 19-20: What are the precursors that microorganisms process to form these compounds?

Page 5 Lines 20-26: This paragraph raises more questions than the answers. How do the authors explain a better correlation of lactic acid with C3-C6 monoacids for night-time samples? Do the lactic acid producing lactobacillus species more active at night? Even they are produced in soil, how is lactic acid emitted from soil to atmosphere? Couldn't lactic acid be a product of atmospheric oxidation processes? The assumption that C1-C6 monoacids and lactic acid are emitted from soil is not valid without providing concrete evidence for direct soil emission data.

Page 5 Lines 29-32: This paragraph does not add useful information to the manuscript without leaf and soil emission data. They can also be produced from atmospheric oxidation processes.

Page 6 Line 15: Is it important for the formation of organic aerosols or is it important for the formation of biogenic SOA marker compounds? It should also be clarified if the acidity is important for enhancing the partitioning of gaseous organic compounds into the particle phase or the acidity is important for the formation of biogenic tracer compounds in the gas or particle phase.

Page 6 Line 17: Aren't formic acid and acetic acid end products rather than intermediate products of oxidation? They may be very stable intermediate products if you think CO2 as the end product though.

[Figure]

Page 6 Lines 18-19: This sentence is confusing. Do the authors mean lower pH values of aerosol than pKa values of formic acid (∼3.7) and acetic acid (∼4.7)? If this is the case, wouldn't formic acid and acetic acid go out of the particle phase, leading to lower particle phase concentrations of both species? If the acidity enhances the formation of small organic acids from BVOCs, what are the processes involved in the formation? Do BVOCs get oxidized directly by particle phase acids on the particle surface?

Page 6 Lines 19-24: What about the boundary layer height and temperature? It is often observed that the concentrations of particle phase organic compounds are higher in nighttime samples than daytime samples due to shallower boundary layer height and lower temperature.

Page 6 Line 25: Which bacterium is it (species, strain, etc)? What conditions does it need to produce isopentanoic acid? Is it atmospherically relevant condition? I am sure that not all bacteria are isopetanoic acid producers.

Page 6 Lines 25-29: This paragraph does not add up. Do the authors suggest that lactic acid and isopentanoic acid may be produced by bacteria in soil and emitted to atmosphere because they did not correlate well with particle phase monoacids that might be also produced by soil bacteria? Where are the evidence for particle surface formic acid and acetic acid formation? The data provided in the manuscript is hardly conclusive to suggest the sources or processes involved in their formation.

Page 7 Lines 7-8: I am not sure how the comparison of the vapor pressures and Fp values for these compounds is related to the source of lactic acid and isopentanoic acid. Please clarify this.

Page 7 Line 10: with increasing ambient temperature

Page 7 Line 14: What about the nighttime?

Page 7 Lines 15-18: Please provide LWC data. What were the state of particles? Were particles deliquesced? How does Fp correlate with the total particle phase acid

concentration? It may well be that particle bound organic mass may be more important for the partitioning of monoacids than LWC.

The section 4.3 does not add useful information to the manuscript as the data compared here are only snapshots of monoacid concentrations at each sampling site, and do not take seasonal or temporal variations of monoacid concentrations into account. I suggest the authors removing this section completely.

The summary and conclusion section need to be revised. There are too many speculative statements here rather than evidence based statements.

---

## Author Comment (AC1) · 29 Aug 2018

We revised the manuscript following the reviewers' comments. We prepared Response Letter in which we explained our changes/correction on a point-by-point basis. Please find the response letter and revised MS.

Please also note the supplement to this comment:
https://www.atmos-chem-phys-discuss.net/acp-2018-444/acp-2018-444-AC1-supplement.zip

---

## Author Response (AR1)

**Authors' Responses to reviewers**

We appreciate the helpful comments made by reviewers.

Following the two reviewer's comments, we have removed Figure 7. Figure numbers were reorganized.

We have changed the configuration of section 3 (Results).

Below, we indicate our point-to-point responses in blue.

**Reviewer 1:**

*Manuscript describes the results from the measurements of low molecular weight acids in the forest air in Japan. These acids have effects on atmospheric chemistry, acidity and clouds, but their sources and concentrations in the air are poorly known. Even with very limited data set, study gives new knowledge on these poorly known compounds.*

*However, major improvements on interpreting the results is needed.*

*Major comments:*

*Comment 1: It would be very interesting to see how big fraction of the particulate mass these acids comprise. Could you add some information on particle mass?*

**Response:**

Thanks for the comment. Unfortunately, we did not measure the aerosol mass. However, we confirmed the presence of a relationship between particle-phase fractions ($F_p$) and mass concentrations of total LMW monoacids in particle phase, except for $C_3$, $C_9$, and $C_{10}$ monoacids. We have added the following sentences in the revised manuscript.

"$F_p$ of LMW monoacids showed positive correlations with mass concentrations of total LMW monoacids in particle phase ($r^2 = 0.24$–$0.46$), except for $C_3$, $C_9$, and $C_{10}$. Although we did not measure the aerosol mass, gaseous LMW monoacids in the forest atmosphere may be adsorbed on the existing particles." Please see page 6, lines 29-31.

*Comment 2: More discussion on other possible sources are needed. For example production in the air in gas phase may be very important and also transport from anthropogenic or other sources.*

**Response:**

We have checked backward air mass trajectories at a level of 300 m. We have added the following sentences in experimental and result sections.

"We calculated seven-day air mass back trajectories at a height of 300 m above sea level using the Meteorological Data Explorer (METEX) provided by the National Institute for Environmental Studies (http://db.cger. nies.go.jp/metex/index.html)." Please see page 3, lines 31-33.

"This forest site is located a few kilometers south of Sapporo city. The dominant wind direction

was from east and south throughout the sampling period. We compared the concentrations of individual monoacids together with nss-$SO_4^{2-}$: an anthropogenic aerosol tracer to evaluate the influence of anthropogenic air mass transport from urban area. We confirmed that individual monoacids in both gas and particle phases did not show a correlation with nss-$SO_4^{2-}$ ($r^2 < 0.14$). The majority of sampling air was not affected by urban area. In addition, Figure 5 shows seven-day air mass back trajectories (300 m a.s.l.) for the study period from 28 June to 8 July at the sampling site. Most of the air masses passed through the Pacific Ocean during the measurement period, except for 28 June. This result may suggest that the air masses arriving at the forest site are not affected by the outflows from East Asia and far East of Russia." Please see page 4, lines 19-26.

[Figure]

Figure 5. Seven-day air mass back trajectories at a height of 300 m a.s.l. during the sampling period.

*Comment 3: Please discuss also on lifetime of these compounds and background levels. For example for acetic acid lifetime in the air is around two week while for decanoic acid only few hours.*

**Response:**

Based on the suggestion, we have added the following sentences in the revised manuscript.

"Concentrations of formic and acetic acids in gas phase were higher than those of $C_3$-$C_{10}$ monoacids. The rate constants of gaseous formic, acetic, and propionic acids with OH radicals were 0.74, 1.2, and $2.0 \times 10^{-12}$ $cm^3$ molecule$^{-1}$ s$^{-1}$, respectively (Dagaut et al., 1988). The rate constants of LMW monoacids may increase with an increase in carbon numbers of monoacids. Formic and acetic acids are more stable than $C_3$-$C_{10}$ due to more hydrogen atoms in $C_3$-$C_{10}$ acids to react with OH radicals, leading to lower concentrations of $C_3$-$C_{10}$ monoacids in the atmosphere." Please see page 4, lines 29-33.

"The lifetimes of formic and acetic acids in gas phase are estimated to be 25 and 10 days, respectively (Paulot et al., 2011). These acids can be long range transported in the atmosphere. In gas phase, formic and acetic acids showed positive correlations with short-lived monoacids ($C_3$-$C_6$ monoacids) (day: $r^2$ = 0.17-0.89, night: $r^2$ = 0.14-0.65)." Please see page, 4 lines 36-39.

*Comment 4: Page 5, line 24-25: From only ambient air concentrations you cannot state that some are emitted from forest floor and soil microorganisms. Or do you have data on microorganisms? If so, please show it in manuscript. There are several different sources and sinks possibly affecting the concentration levels and for some compounds even background levels may be relatively high.*

**Response:**

We modified the sentences in the revised manuscript as follows.

"Branched chain monoacids including isobutylic acid are known as common metabolites of bacteria (e.g., *Bacteroides distasonis*) and fungi in soils (Uta et al., 2012 and references therein). LMW monoacids such as acetic and propionic acids can be produced by microbiological processes (Uta et al., 2012). In addition, exudation of organic acids is known to occur in vascular plants, mainly from roots (Curl, 1982). Shen et al. (1996) reported that formic, acetic, and propionic acids are contained in forest soil and rhizosphere soil. However, the possible contribution from the forest floor cannot be evaluated in the present study." Please see page 4, line 42 - page 5, line 5.

*Comment 5: Page 5 and 6, lines 30 and 1-2: Even ambient air concentrations of monoterpenes, with clear temperature and light dependent emissions, do not most often show direct correlation with temperature since there are several processes affecting on ambient concentration levels in the air (e.g. mixing layer height, vertical wind, background levels and reactivity). Therefore I recommend you to remove this statement on contribution of soil being higher than forest canopy.*

**Response:**

Deleted as suggested.

*Comment 6: Page 6, lines 10-18: This paragraph is very unclear. It was not clear to me, when you discuss on gas phase and when particulate. I would expect that lots of formic and acetic acids are formed in the gas phase reactions of BVOCs and then these formed acids partition into particulate phase. Of course some additional could then be formed also in particles or their surfaces.*

**Response:**

We deleted the corresponding sentences and Figure 7 in the manuscript. Instead, we have added sentences in the revised manuscript. Please see our response to comment 7 as given

below.

*Comment 7: Page 6, line 23-24: BVOCs with double bonds do react in the air with $O_3$ and $NO_3$ also during the night. But after first reactions they have lost the double bonds and continue reactions with OH radicals, which are formed only during the day. Double bonds from BVOCs are lost well-before they form formic and acetic acid and therefore I expect that formic and acetic acids are formed in the reactions of other VOCs only during the day. I would expect that due to partitioning between gas and aerosol phase more is detected in particulate phase during colder the nights and then during the day when temperature increases they are evaporated. You should add discussion also on this.*

**Response:**

Thanks for the comment. We reorganized this section by deleting the sentence, "Formic and acetic acids are later-generation products of the oxidation of BVOCs (e.g., Ervens et al., 2008). BVOCs or their earlier-generation products may be continuously oxidized at night in the forest atmosphere."

Instead, we have added few sentences as follows.

"Gas/particle partitioning of formic and acetic acids are discussed in section 4.2." Please see page 5, line 26.

"Although $F_p$ of LMW monoacids did not show a correlation with ambient temperature in nighttime ($r^2 < 0.16$), except for propionic acid ($r^2 = 0.31$), we found average $F_p$ of LMW monoacids in nighttime were higher than those in daytime (Table 1)." Please see page, 5 line 42 - page, 6 line 2.

"We found that $F_p$ of formic and acetic acids increase with increasing RH ($C_1$: $r^2 = 0.30$, $C_2$: $r^2 = 0.55$) (Figure 9), whereas other LMW monoacids did not show a correlation with RH ($r^2 < 0.20$), except for butyric acid ($r^2 = 0.55$) in daytime. Although $F_p$ of LMW monoacids did not show correlation with RH in nighttime ($r^2 < 0.15$), we found that average $F_p$ of LMW monoacids in nighttime were higher than those in daytime (Table 1). Al-Hosney et al. (2005) and Prince et al. (2008) reported that the uptake of formic and acetic acids by $CaCO_3$ can be enhanced by higher RH ($C_1$: RH > 62%, $C_2$: RH > 53%). In addition, liquid water contents (LWC) of aerosols were estimated using ISORROPIA-II model (Fountoukis and Nenes, 2007) with the data of inorganic ions and meteorological parameters. The aerosol LWC ranged from 1.4 to 14.6 µg m$^{-3}$ (av. 6.4 µg m$^{-3}$). Although $F_p$ of LMW monoacids did not show strong correlations with LWC ($r^2 < 0.24$), strong positive correlations were found between RH and aerosol LWC in daytime ($r^2 = 0.47$) and in nighttime ($r^2 = 0.74$). Higher RH may enhance the partitioning of gaseous formic and acetic acids to aerosol phase as a result of the condensation of water vapour on aerosol particles. Our results suggest that higher temperature depresses a

transfer of gaseous formic and acetic acids to aerosol phase and higher RH enhances the partitioning of gaseous formic and acetic acids to aerosol phase in the forest atmosphere." Please see page 6, lines 5-17.

We have added new Figure 9 as below.

[Figure]

Figure 9. Particle-phase fractions ($F_p$) of formic and acetic acids against relative humidity.

*Comment 8: Page 7, line 4-8: If lactic acid is more polar and more water soluble it will partition more to the particulate phase. Discuss also on this possibility.*

**Response:**

Thanks for the comment. Based on the comment, we added the following sentences.

"Although lactic acid is highly water-soluble, $F_p$ of lactic acid did not show a clear correlation with relative humidity. High relative humidity (average: 87%) may be involved with the large $F_p$ of lactic acid." Please see page 6, lines, 21-22.

We deleted the following sentence; "These results suggest that particulate lactic and isopentanoic acids may be derived from primary sources in forest ecosystem via microbial activity."

*Comment 9: Page 7, line 13. Effect of temperature applies also during nighttime.*

**Response:**

Based on comment 7, we already added the sentence in the revised manuscript. Please see our response to comment 7.

*Comment 10: More discussion on also sinks and lifetimes and effect of local chemistry is needed.*

**Response:**

Based on comment 3, we have added and modified the sentences in the revised manuscript. Please see our response to comment 3.

*Comment 11: Page 8, line 29. Add references on branched chain monoacids being an indicator*

*of soil microorganism processed.*

**Response:**

We have added a reference in the revised manuscript as below.

"Branched chain monoacids can be used as an indicator of soil microorganism processes (Uta et al., 2012 and references therein)." Please see page 7, lines 1-2.

*Comment 12: Page 8, line 30: Based on the data, it cannot be stated that microorganisms are source of these acids. They can as well be produced in the local reactions of other emitted compounds.*

**Response:**

Following the comment, the sentence was modified as below.

"Many kinds of LMW monoacids are exuded by plant roots (Curl, 1982) and derived from soil microorganism (Uta et al., 2012). We suggest that forest floor is another important source of gaseous LMW monoacids." Please see page 7, lines 2-4.

*Comment 13: Page 8, line 32: Do you have any data on microbial activity? If not, remove this statement.*

**Response:**

Deleted as suggested.

*Comment 14: Page 9, line 3: Based on the data shown here, this cannot be stated. Maybe they give some indication that soil could be a source, but it cannot be stated.*

**Response:**

We deleted the following sentences; "Forest floor including soil microorganisms contributes to the emissions of gaseous and particulate LMW monoacids. Our results may be useful to improve the understanding of organic aerosol formation in the forest."

*Comment 15: Even with low concentrations C4-C10 acids can be very important and have higher local effect and higher emissions/sources that you would think by their concentrations. This is because their reactivity and lifetimes in the air are much shorter. Especially for these acids data on ambient concentrations and sources are very limited. Please, add more discussion also on these acids.*

**Response:**

Please see our response to comment 3.

*Comment 16: Could you add a table on the comparison with other studies?*

**Response:**

Based on the reviewer 2' comment 15, we deleted section 4.3.

*Minor comments:*

*Comment 17: Language should be checked by some native speaker.*

**Response:**

We improved English as best as we could.

*Comment 18: Page 4 lines 15-18. Frist you tell that isopentanoic acid is most dominant in particles and then nonanoic acid. Please, correct these sentences.*

**Response:**

We deleted the following sentence; "Nonanoic acid ($C_9$) is the dominant species (1.5–25 ng m$^{-3}$, mean: 11 ng m$^{-3}$) in particle phase, except for formic and acetic acids (Figure 3)."

*Comment 19: Page 4 line 25. What temporal trend? Please explain more clearly.*

**Response:**

We are sorry for the unclearness. We have rephrased the sentences as below.

"Day-to-day variations in monoacids detected did not show a clear diurnal trend." Please see page 4, lines 4-5.

*Comment 20: Fig. 4. It is very hard to deviate between the days. Could you add for example some dashed lines between the days?*

**Response:**

We modified Figure 4 as follows. Please see the revised figure.

[Figure]

Figure 4. Diurnal variations in the concentrations of major monocarboxylic acids in gas (open circle) and particle (solid diamond) phases and inorganic ions (nss-$SO_4^{2-}$ and $NH_4^+$). Day: D, Night: N.

**Reviewer 2:**

*The authors of this paper report the concentrations of low molecular weight monocarboxylic acids in the gas and particle phase samples collected at a deciduous broad leaf forest site near Sapporo, northern Japan. They claim that acetic acid, isopentanoic acid, and lactic acid are the major particle phase monoacids at the site and they claim that isopentanoic acid and lactic acid originate from soil bacteria activities because they correlate well with gas phase isobutyric acid that may be produced by soil bacteria. While some of the data presented here may be interesting for people working in this field, the paper has major flaws as there are too many speculative statements that do not stand up to scrutiny. I recommend major revision before this paper could be published in ACP.*

*Comment 1: Page 5 Line 12: It is not clear to me why the authors chose sulfate ion as an anthropogenic tracer for local urban emissions. Is the total sulfate ion or is it nss-sulfate? How does the authors exclude the influence of long range transport from China or Russia? This should be clarified.*

**Response:**

Thanks for the comment.

We calculated non-sea-salt $SO_4^{2-}$ and added the following sentences/phrases in the revised manuscript.

"Concentrations of non-sea-salt $SO_4^{2-}$ [nss-$SO_4^{2-}$] is calculated by the following equation:

[nss-$SO_4^{2-}$] = [$SO_4^{2-}$] – 0.25×[$Na^+$],

where [$SO_4^{2-}$] and [$Na^+$] are concentrations of total $SO_4^{2-}$ and $Na^+$, respectively (Duce et al., 1983, Berg and Winchester, 1978)." Please see page 3, lines 26-30.

"Nss-$SO_4^{2-}$ (mean: 2240 ng m$^{-3}$) is major anion …" Please see page 4, line 16.

"nss-$SO_4^{2-}$". Please see page 4, line 20.

"nss-$SO_4^{2-}$ ($r^2 < 0.14$)". Please see page 4, line 22.

We also checked backward air mass trajectories at a level of 300 m and added the following sentences in experimental and result sections.

"We calculated seven-day air mass back trajectories at a height of 300 m above sea level using the Meteorological Data Explorer (METEX) provided by the National Institute for Environmental Studies (http://db.cger. nies.go.jp/metex/index.html)." Please see page 3, lines 31-33.

"This forest site is located a few kilometers south of Sapporo city. The dominant wind direction was from east and south throughout the sampling period. We compared the concentrations of

individual monoacids together with nss-SO$_4^{2-}$: an anthropogenic aerosol tracer to evaluate the influence of anthropogenic air mass transport from urban area. We confirmed that individual monoacids in both gas and particle phases did not show a correlation with nss-SO$_4^{2-}$ (r$^2$ < 0.14). The majority of sampling air was not affected by urban area. In addition, Figure 5 shows seven-day air mass back trajectories (300 m a.s.l.) for the study period from 28 June to 8 July at the sampling site. Most of the air masses passed through the Pacific Ocean during the measurement period, except for 28 June. This result may suggest that the air masses arriving at the forest site are not affected by the outflows from East Asia and far East of Russia." Please see page 4, lines 19-26.

[Figure]

Figure 5. Seven-day air mass back trajectories at a height of 300 m a.s.l. during the sampling period.

*Comment 2: Page 5 Lines 19-20: What are the precursors that microorganisms process to form these compounds?*
**Response:**
Following the comment, we added the following sentence with reference; "Branched chain monoacids including isobutylic acid are known as common metabolites of bacteria (e.g., *Bacteroides distasonis*) and fungi in soils (Uta et al., 2012 and references therein)." Please see page 4, line 42 - page 5, line 2.

*Comment 3: Page 5 Lines 20-26: This paragraph raises more questions than the answers. How do the authors explain a better correlation of lactic acid with C3-C6 monoacids for nighttime samples? Do the lactic acid producing lactobacillus species more active at night? Even they are produced in soil, how is lactic acid emitted from soil to atmosphere? Couldn't lactic acid be a product of atmospheric oxidation processes? The assumption that C1-C6 monoacids and*

*lactic acid are emitted from soil is not valid without providing concrete evidence for direct soil emission data.*

**Response:**

We deleted the corresponding sentences. Instead, we have added the following sentences in the revised manuscript.

"LMW monoacids such as acetic and propionic acids can be produced by microbiological processes (Uta et al., 2012). In addition, exudation of organic acids is known to occur in vascular plants, mainly from roots (Curl, 1982). Shen et al. (1996) reported that formic, acetic, and propionic acids are contained in forest soil and rhizosphere soil. However, the possible contribution from the forest floor cannot be evaluated in the present study. Variety of emission sources of $C_1$–$C_6$ monoacids may exist in forest soil. We suggest that gaseous $C_1$–$C_6$ monoacids are emitted from the forest floor where soil microorganisms and plant roots contribute to the emissions of gaseous $C_1$–$C_6$ monoacids.

Gaseous lactic acid ($hC_3$) in daytime did not show positive correlations with $C_1$–$C_6$ monoacids ($r^2 < 0.004$), whereas lactic acid in nighttime show positive correlations with $C_3$–$C_6$ ($r^2 = 0.45$–$0.65$) although they were rather scattered. Bacteria (*lactobacillus*) are known to produce lactic acid (Cabredo et al., 2009). Lactic acid can also be produced by the oxidation of isoprene with ozone (Nguyen et al., 2010). We suggest that major portion of $C_1$-$C_6$ acids were emitted within the forest floor." Please see page 5 lines, 2-11.

*Comment 4: Page 5 Lines 29-32: This paragraph does not add useful information to the manuscript without leaf and soil emission data. They can also be produced from atmospheric oxidation processes.*

**Response:**

Taking account of the comment, we deleted those sentences.

*Comment 5: Page 6 Line 15: Is it important for the formation of organic aerosols or is it important for the formation of biogenic SOA marker compounds? It should also be clarified if the acidity is important for enhancing the partitioning of gaseous organic compounds into the particle phase or the acidity is important for the formation of biogenic tracer compounds in the gas or particle phase.*

**Response:**

Following the comments, we deleted the following sentences; "Particulate formic and acetic acids in daytime negatively correlated with pH of the water extracts from aerosol samples ($C_1$: $r^2 = 0.35$, $C_2$: $r^2 = 0.43$) (Figure 7). Acidity of aerosol particles can enhance the formation of SOA via organic precursors and subsequent heterogeneous reactions in aerosol liquid phase (e.g., Jang et al., 2002; Pathak et al., 2011). Secondary formation of formic and acetic acids is associated with photo-oxidation of biogenic volatile organic compounds (BVOCs) such as isoprene and monoterpene (Ervens et al., 2008; Kawamura et al., unpublished data). Based on

the field measurement, aerosol acidity is important for the formation of organic aerosols such as isoprene SOA tracers, α-pinene SOA tracers and dicarboxylic acids (oxalic acid) from BVOCs in the forest atmosphere (Mochizuki et al., 2015; 2017b). Formic and acetic acids are intermediate products in the complex heterogeneous oxidation of BVOCs. Lower pH may promote the secondary formation of formic and acetic acids from BVOCs in the forest atmosphere (Kawamura et al., unpublished data)."

*Comment 6: Page 6 Line 17: Aren't formic acid and acetic acid end products rather than intermediate products of oxidation? They may be very stable intermediate products if you think CO$_2$ as the end product though.*

**Response:**

Thanks for your comment. The lifetimes of formic and acetic acids in gas phase are estimated to be 25 and 10 days, respectively (Paulot et al., 2011). Formic and acetic acids are stable in the atmosphere. By taking the comment, we modified the sentence in the revised MS. Please see page 5 lines 24-25.

*Comment 7: Page 6 Lines 18-19: This sentence is confusing. Do the authors mean lower pH values of aerosol than pKa values of formic acid (3.7) and acetic acid (4.7)? If this is the case, wouldn't formic acid and acetic acid go out of the particle phase, leading to lower particle phase concentrations of both species? If the acidity enhances the formation of small organic acids from BVOCs, what are the processes involved in the formation? Do BVOCs get oxidized directly by particle phase acids on the particle surface?*

**Response:**

Based on the reviewer's comment 5, we deleted the relevant sentences.

*Comment 8: Page 6 Lines 19-24: What about the boundary layer height and temperature? It is often observed that the concentrations of particle phase organic compounds are higher in nighttime samples than daytime samples due to shallower boundary layer height and lower temperature.*

**Response:**

Although we do not have the information on boundary layer height, one sentence was added on the boundary layer height following the reviewer's comment.

"The higher concentrations in nighttime may be associated with shallower planetary boundary layer, which can accumulate organic acids near the ground surface." Please see page 5, lines 20-21.

Ambient temperature and relative humidity are important factors to control the gas/particle partitioning of formic and acetic acids. Following the reviewer's comment, we have added the following sentences.

"Although $F_p$ of LMW monoacids did not show a correlation with ambient temperature in nighttime ($r^2 < 0.16$), except for propionic acid ($r^2 = 0.31$), we found average $F_p$ of LMW monoacids in nighttime were higher than those in daytime (Table 1)." Please see page 5, line 42 - page 6, line 2.

*Comment 9: Page 6 Line 25: Which bacterium is it (species, strain, etc)? What conditions does it need to produce isopentanoic acid? Is it atmospherically relevant condition? I am sure that not all bacteria are isopetanoic acid producers.*

**Response:**

Based on the comment, we rephrased the sentence as follows.

The sentence "Isopentanoic acid can be produced by bacteria (Allison, 1978)." has been changed to "Isopentanoic acid can be produced by bacteria such as *Clostridium* spp. and *Bacteroides* spp. (Uta et al., 2012 and references therein). These microorganisms live in soil." Please see page 5, lines 27-28.

*Comment 10: Page 6 Lines 25-29: This paragraph does not add up. Do the authors suggest that lactic acid and isopentanoic acid may be produced by bacteria in soil and emitted to atmosphere because they did not correlate well with particle phase monoacids that might be also produced by soil bacteria? Where are the evidence for particle surface formic acid and acetic acid formation? The data provided in the manuscript is hardly conclusive to suggest the sources or processes involved in their formation.*

**Response:**

Thanks for the critical comment. We have modified the sentence in the revised manuscript as:

"We suggest that lactic and isopentanoic acids are linked to the biosynthetic processes, whereas formation processes of $C_1$-$C_6$ acids in particle phase may be different from isopentanoic and lactic acids." Please see page 5, lines 30-31.

*Comment 11: Page 7 Lines 7-8: I am not sure how the comparison of the vapor pressures and Fp values for these compounds is related to the source of lactic acid and isopentanoic acid. Please clarify this.*

**Response:**

Following the suggestion, we deleted the following sentence; "These results suggest that particulate lactic and isopentanoic acids may be derived from primary sources in forest ecosystem via microbial activity."

*Comment 12: Page 7 Line 10: with increasing ambient temperature*

**Response:**

Corrected as suggested. Please see page 7, line 14.

*Comment 13: Page 7 Line 14: What about the nighttime?*

**Response:**

Please see our response to comment 8.

*Comment 14: Page 7 Lines 15-18: Please provide LWC data. What were the state of particles? Were particles deliquesced? How does Fp correlate with the total particle phase acid concentration? It may well be that particle bound organic mass may be more important for the partitioning of monoacids than LWC.*

**Response:**

We added the aerosol liquid water content in the revised MS with additional discussion.

"We found that $F_p$ of formic and acetic acids increase with increasing RH (C$_1$: r$^2$ = 0.30, C$_2$: r$^2$ = 0.55) (Figure 9), whereas other LMW monoacids did not show a correlation with RH (r$^2$ < 0.20), except for butyric acid (r$^2$ = 0.55) in daytime. Although $F_p$ of LMW monoacids did not show correlation with RH in nighttime (r$^2$ < 0.15), we found that average $F_p$ of LMW monoacids in nighttime were higher than those in daytime (Table 1). Al-Hosney et al. (2005) and Prince et al. (2008) reported that the uptake of formic and acetic acids by CaCO$_3$ can be enhanced by higher RH (C$_1$: RH > 62%, C$_2$: RH > 53%). In addition, liquid water contents (LWC) of aerosols were estimated using ISORROPIA-II model (Fountoukis and Nenes, 2007) with the data of inorganic ions and meteorological parameters. The aerosol LWC ranged from 1.4 to 14.6 µg m$^{-3}$ (av. 6.4 µg m$^{-3}$). Although $F_p$ of LMW monoacids did not show strong correlations with LWC (r$^2$ < 0.24), strong positive correlations were found between RH and aerosol LWC in daytime (r$^2$ = 0.47) and in nighttime (r$^2$ = 0.74). Higher RH may enhance the partitioning of gaseous formic and acetic acids to aerosol phase as a result of the condensation of water vapour on aerosol particles. Our results suggest that higher temperature depresses a transfer of gaseous formic and acetic acids to aerosol phase and higher RH enhances the partitioning of gaseous formic and acetic acids to aerosol phase in the forest atmosphere." Please see page 6, lines 5-17.

We have added new Figure 9 as below.

[Figure]

Figure 9. Particle-phase fractions ($F_p$) of formic and acetic acids against relative humidity.

We also added the following sentences in the revised manuscript.

"$F_p$ of LMW monoacids showed positive correlations with mass concentrations of total LMW monoacids in particle phase ($r^2$ = 0.24–0.46), except for $C_3$, $C_9$, and $C_{10}$. Although we did not measure the aerosol mass, gaseous LMW monoacids in the forest atmosphere may be adsorbed on the existing particles." Please see page 6, lines 29-31.

*Comment 15: The section 4.3 does not add useful information to the manuscript as the data compared here are only snapshots of monoacid concentrations at each sampling site, and do not take seasonal or temporal variations of monoacid concentrations into account. I suggest the authors removing this section completely.*

**Response:**

Deleted as suggested.

*Comment 16: The summary and conclusion section need to be revised. There are too many speculative statements here rather than evidence based statements.*

**Response:**

According to the comment, we have revised the summary and conclusions.

[revised manuscript text omitted]

---

## Author Response (AR2)

We appreciate the helpful comments made by Co-Editor and reviewers.

Below, we indicate our point-to-point responses in blue.

**Co-Editor:**

*Comments to the Author:*

*The two reviewer judged the revised manuscript. At least one reviewer still showed great concern about the scientific quality. Particularly the reviewer requests some more concrete evidence that lactic acid is produced by microorganisms, and to argue the possibility of other sources (e.g., photochemistry) more carefully. I find that revision is on the right track but would need to consider if the authors could address to these points.*

*My additional concern is on Figure 9. How to interpret the negative correlation with RH during nighttime? Better not to show regression lines when the correlation is not statistically significant? (also for other figures)*

**Response:**

Based on the reviewer'2 comment, we modified the corresponding sentences in the revised manuscript after the additional lab work on the analysis of forest soil samples for monoacids.

"Gaseous lactic acid in daytime did not show positive correlations with $C_1$–$C_6$ monoacids ($r^2 <$ 0.004), whereas lactic acid in nighttime show positive correlations with $C_3$–$C_6$ ($r^2 = $ 0.45–0.65) although they were rather scattered. Particulate lactic acid also did not show correlations with other LMW monoacids detected in particle phase ($r^2 < 0.17$). Formation processes of $C_1$-$C_6$ acids may be different from hydroxy monoacids." Please see page 5, lines, 11-14.

"In this study, relatively high concentrations of particulate lactic and isopentanoic acids were observed. Lactic acid is primary produced by Bacteria (*lactobacillus*) (Cabredo et al., 2009) and from the plant tissues (Raja et al., 2008). Lactic acid can also be produced by the oxidation of isoprene with ozone in (Nguyen et al., 2010). Isopentanoic acid can be produced by bacteria such as *Clostridium* spp. and *Bacteroides* spp. (Uta et al., 2012 and references therein). A positive correlation was observed between lactic acid and isopentanoic acids in particle phase ($r^2 = 0.98$). We confirmed that lactic acid is abundantly present in forest soil (1860 ng $g_{wet\ soil}^{-1}$) but isopentanoic acid is not (unpublished data). Lactic and isopentanoic acids may be linked to the biosynthetic processes in the forest soil system as well." Please see page 5, lines 22-28.

Based on the suggestion, regression line was deleted when the correlation is not statistically significant (please see Figures 6, 7, 8, and 9). We have added the following sentence in figure captions (Figures 6, 7, 8, and 9).

The sentence "The coefficient of determination shows that regression line is statistically significant ($p < 0.05$)." Please see Figures 6, 7, and 8.

The sentence "The coefficient of determination shows that regression line is statistically significant ($p < 0.1$)." Please see Figure 9.

**Reviewer 1**

*Authors gave sufficient answers to reviewer comments and improved their manuscript based on them. However, I still have some minor concerns listed below:*

*Comment 1. I would think that $SO_4$ is not good tracer for urban air, since general traffic is not usually emitting it. $SO_4$ is mainly emitted by ships and certain industry. Please, correct that. Could you use for example $NO_x$ as an urban traffic tracer?*

**Response:**

$NO_3^-$ is good tracer for urban air. We have modified the sentences in the revised manuscript as follows.

"Concentrations of LMW monoacids did not correlate with nss-$SO_4^{2-}$ and $NO_3^-$ that are used as anthropogenic tracers, indicating that LMW monoacids are derived from the local sources within the forest ecosystem." Please see page 1, lines 16-18.

"We compared the concentrations of individual monoacids together with nss-$SO_4^{2-}$ and $NO_3^-$: anthropogenic aerosol tracers to evaluate the influence of anthropogenic air mass transport from urban area. We confirmed that individual monoacids in both gas and particle phases did not show correlation with nss-$SO_4^{2-}$ ($r^2 < 0.14$) and $NO_3^-$ ($r^2 < 0.11$)." Please see page 4, lines 20-23.

"Concentrations of LMW monoacids in gas and particle phases did not correlate with nss-$SO_4^{2-}$ and $NO_3^-$ (anthropogenic tracers)." Please see page 7, lines 2-3.

*Comment 2. Page 4, line 13: Do you define somewhere, what is short-chain acids and what is long-chain?*

**Response:**

Following the comment, the sentence was modified as below.

"Formic ($C_1$) and acetic ($C_2$) acids are largely present in gas phase." Please see page 4, line 12.

*Comment 3. Check the values in Table 1. At least night time values and $F_p$ for Lactic acid cannot be correct.*

**Response:**

Individual values of $F_p$ were first calculated by each measurement data and then average values of $F_p$ were calculated by individual values of $F_p$. Values of $F_p$ (Table 1) are different from $F_p$ calculated by average concentrations of gaseous and particulate monoacids.

*Comment 4. Page 5, lines 6 and 7: You claim that 'We suggest that gaseous $C_1$–$C_6$ monoacids are*

*emitted from the forest floor where soil microorganisms and plant roots contribute to the emissions of gaseous $C_1$–$C_6$ monoacids'. Based on you data you cannot make this strong conclusion. They can also be produced for example from the reactions of BVOCs.*

**Response:**

Following the comment, the sentence was modified as below.

"Although we detected formic, acetic, propionic, and isobutylic acids in forest soil (Kunwar et al., unpublished data, 2018), a quantitative contribution from the forest floor cannot be evaluated in the present study. On the other hand, photo-oxidation of BVOCs such as isoprene and monoterpenes are major sources of formic and acetic acids in the atmosphere (Paulot et al., 2011). Our results suggest that forest floor may be a source of gaseous $C_1$–$C_6$ monoacids to the atmosphere." Please see page 5, lines 7-10.

We have added the following sentence in abstract.

"Forest soil may be a source of gaseous $C_1$–$C_6$ monoacids in the forest atmosphere." Please see page 1, lines 19-20.

*Comment 5. Page 5, line: You claim that 'We suggest that major portion of $C_1$-$C_6$ acids were emitted within the forest floor'. How do you know that it is not from isoprene oxidation or oxidation of some other VOCs? Do not make this strong conclusion.*

**Response:**

Based on comment 4, we have modified the sentences in the revised manuscript. Please see our response to comment 4.

**Reviewer 2**

*General comments:*

*I do understand that the authors cannot provide evidence for the emission of organic acids from the forest soil in this study. Under such circumstances, I expect the authors to provide strong convincing supporting data (for example, soil analysis data, laboratory bacterial experiment etc.) to strengthen the authors' claim. This is especially important when the authors propose a new source of marker compounds. In this revised manuscript the authors failed to do so, and they only provided a small number of biology related references to suggest microorganisms as a source of these acids in the atmosphere. Based on the comments during the discussion phase, and this comments that are substantiated below, I am not able to recommend this manuscript for publication in ACP.*

*Comment 1: Abstract*

*(Revised) Page 1 line 18: Is lactic acid really produced by microorganisms? Do the authors have concrete evidence for this? Correlation does not imply causation.*

**Response:**

Following the comment, we deleted the following sentence in abstract and conclusion.

The sentence: "Isopentanoic acid in particle phase showed a positive correlation with lactic acid ($r^2 = 0.98$), which is produced by soil microbes."

We have added the following sentence in abstract.

"Forest soil may be a source of gaseous $C_1$–$C_6$ monoacids to the atmosphere." Please see page 1, lines 19-20.

In addition, we modified the corresponding sentences in the revised manuscript.

"Gaseous lactic acid in daytime did not show positive correlations with $C_1$–$C_6$ monoacids ($r^2 <$ 0.004), whereas lactic acid in nighttime show positive correlations with $C_3$–$C_6$ ($r^2 = 0.45$–0.65) although they were rather scattered. Particulate lactic acid also did not show correlations with other LMW monoacids detected in particle phase ($r^2 < 0.17$). Formation processes of $C_1$-$C_6$ acids may be different from hydroxy monoacids." Please see page 5, lines 11-14.

"In this study, relatively high concentrations of particulate lactic and isopentanoic acids were observed. Lactic acid is primary produced by Bacteria (*lactobacillus*) (Cabredo et al., 2009) and from the plant tissues (Raja et al., 2008). Lactic acid can also be produced by the oxidation of isoprene with ozone in (Nguyen et al., 2010). Isopentanoic acid can be produced by bacteria such as *Clostridium* spp. and *Bacteroides* spp. (Uta et al., 2012 and references therein). A positive

correlation was observed between lactic acid and isopentanoic acids in particle phase ($r^2 = 0.98$). We confirmed that lactic acid is abundantly present in forest soil (1860 ng $g_{wet\ soil}^{-1}$) but isopentanoic acid is not (unpublished data). Lactic and isopentanoic acids may be linked to the biosynthetic processes in the forest soil system as well." Please see page 5, lines 22-28.

*Discussion*

*Comment 2: (Revised) Page 4 lines 33: This sentence does not make sense. Do the authors mean that OH oxidation is the sole reason for their different concentrations in the atmosphere?*

**Response:**

Following the comment, the sentence was modified as below.

"Formic and acetic acids are stable, leading to higher concentrations of formic and acetic acids in the atmosphere. In addition, we measured LMW monoacids in soil sample (surface ~3 cm) collected at a broad-leaf forest from Chubu University campus in Central Japan on October 31, 2018. LMW normal ($C_1$–$C_4$, $C_7$–$C_{10}$), branched chain ($iC_4$), and hydroxyl (lactic and glycolic) monoacids were detected. Concentrations of formic (7400 ng $g_{wet\ soil}^{-1}$) and acetic (4260 ng $g_{wet\ soil}^{-1}$) acids were significantly higher than that of other detected monoacids (~ 1800 ng $g_{wet\ soil}^{-1}$) (Kunwar et al., unpublished data, 2018). Major portion of monoacids in the forest atmosphere is similar to that in forest soil sample, suggesting forest soil is a source of LMW monoacids as well." Please see page 4, lines 35-41.

*Comment 3: (Revised) Page 4 line 30: These OH rate constants aren't typically accepted values for these compounds. Please refer to NIST Kinetic Database for the list of more widely accepted gas phase OH rate constants of these compounds.*

**Response:**

Following the comment, the sentences were modified as below.

"The rate constants of gaseous formic, acetic, propionic, butyric, and isobutyric acids with OH radicals are 0.45, 0.67, 1.20, 1.79, and 2.06×$10^{-12}$ $cm^3$ molecule$^{-1}$ s$^{-1}$, respectively (provided by NIST Chemical Kinetics Database). The rate constants of LMW monoacids with oxidant (OH radicals and ozone) may increase with an increase in carbon numbers of monoacids." Please see page 4, lines 32-35.

*Comment 4: (Revised) Page 4 line 39: What do you mean by short lived mono acid? What are the lifetimes of C3-C6 acids? From the gas phase OH rate constant of butyric acid ($1.79E^{-12}$ $cm^3$/molecule s), larger acids do not seem to be much short-lived when OH oxidation is solely considered.*

**Response:**

The lifetimes of monocarboxylic acids (more large acids) may depend on not only OH radical oxidation but also $O_3$ oxidation. The lifetimes of monoacids (> $C_5$ monoacids) have not been reported. Based on comments 2 and 3, we have modified the sentences in the revised manuscript. Please see our responses to comments 2 and 3.

*Comment 5: (Revised) Page 5 lines 1-7: A small number of references to related microbial activities and plant physiology aren't enough to claim a source of marker compounds in atmospheric samples, especially when environmental variables and biological activities can play significant roles in the emissions and transformations of precursor compounds. The authors may not have not taken soil samples during the campaign, but they can always go back later to get some samples, try to understand soil bacteria, or analyze extractable soil organics.*

**Response:**

As suggested, we measured LMW monoacids in soil sample (surface ~3 cm) collected at a broad-leaf forest from Chubu University in Central Japan on October 31, 2018. LMW normal ($C_1$–$C_4$, $C_7$–$C_{10}$), branched chain ($iC_4$), and hydroxyl (lactic and glycolic) monoacids were detected in the soil sample. We modified the corresponding sentences and added the sentences in the revised manuscript.

"In addition, we measured LMW monoacids in soil sample (surface ~3 cm) collected at a broad-leaf forest from Chubu University campus in Central Japan on October 31, 2018. LMW normal ($C_1$–$C_4$, $C_7$–$C_{10}$), branched chain ($iC_4$), and hydroxyl (lactic and glycolic) monoacids were detected. Concentrations of formic (7400 ng $g_{wet\ soil}^{-1}$) and acetic (4260 ng $g_{wet\ soil}^{-1}$) acids were significantly higher than that of other detected monoacids (~ 1800 ng $g_{wet\ soil}^{-1}$) (Kunwar et al., unpublished data, 2018). Major portion of monoacids in the forest atmosphere is similar to that in forest soil sample, suggesting forest soil is a source of LMW monoacids as well." Please see page 4, lines 36-41.

"Although we detected formic, acetic, propionic, and isobutylic acids in forest soil (unpublished data), a quantitative contribution from the forest floor cannot be evaluated in the present study." Please see page 5, lines 7-8.

"We confirmed that lactic acid is abundantly present in forest soil (1860 ng $g_{wet\ soil}^{-1}$) but isopentanoic acid is not (unpublished data)." Please see page 5, lines 26-27.

*Comment 6: (Revised) Page 5 lines 24: "somewhat end products" does not makes sense. It is either "end product" or "intermediate".*

**Response:**

Corrected as suggested. Please see page 5, line 41.

Reference added.

Raja, S., Raghunathan, R., Yu, X. Y., Lee, T., Chen, J., Kommalapati, R. R., Murugesan, K., Shen, X., Qingzhong, Y., Valsaraj, K. T., and Collett Jr., J. L.: Fog chemistry in the Texas-Louisiana Gulf Coast corridor, Atmos. Environ., 42, 2048-2061, 2008.

---

## Author Response (AR3)

**Authors' Response to Co-Editor**

*After the revision, I need to say that overall integrity was reduced. I will reconsider after major revisions.*
**Response:**
We appreciate the critical but helpful comments. Please find our point-to-point responses as below.

*1. Two major flaws: First, monoacids will not react with $O_3$. Second, the new sentence in Abstract that "concentrations of LMW monoacids did not correlate with nss-$SO_4^{2-}$ and $NO_3^-$, indicating that LMW monoacids are derived from the local sources" has a logical flaw. The fact indicates that they are not likely from urban sources, and never indicate about the local sources.*
**Response:**
We confirmed that gaseous LMW monoacids will not react with $O_3$.

Following the comment, the sentence was modified as below.
"Concentrations of LMW monoacids did not show correlations with anthropogenic tracers such as nss-$SO_4^{2-}$ and $NO_3^-$, indicating that anthropogenic contribution is not important." Please see page 1, lines 17–18.

*2. Uta et al. (2012) was cited many times in text as important source of information for this manuscript, but it did not appear in the reference list.*
**Response:**
We made a mistake in the author name.
We have replaced "Uta et al., 2012" by "Effmert et al., 2012".

We also added the following papers in the reference section.
Kawamura, K. and Gagosian, R. B.: Implication of ω-oxocarboxylic acids in the remote marine atmosphere for photooxidation of unsaturated fatty acids, Nature, 325, 330-332, 1987.
Tsai, Y. I. and Kuo, S. C.: Contributions of low molecular weight carboxylic acids to aerosols and wet deposition in a natural subtropical broad-leaved forest environment, Atmos. Environ., 81, 270-279, 2013.

*3. It was good to include soil analysis, but the sampling was made at a different site, for which no justification was provided. Detection of high concentrations of lactic acid, $C_1$ and $C_2$ monoacids in soil supported the idea that they originate from soil or bacterial processes, but the fact that gaseous lactic acid did not show positive correlation with $C_1$-$C_6$ monoacids (page 5, lines 11-12) did not support the idea. Further, the authors claim first that branched chain monoacids including $iC_4$ (and $iC_5$, isopentanoic acid) are known as common metabolites of bacteria (page 5, line 3) and regarded them as a tracer from that emission sector. But they did not show up in high concentrations from soil analysis, denying their assumption. Overall my impression is that the soil analysis introduced more puzzle. Analysis of soil from the same site as atmospheric measurements were made is important.*
**Response:**
Following the comment, we modified the corresponding sentences and added few sentences in the revised manuscript, as follows.
"Although we did not collect a forest soil sample from Sapporo during the air-sampling period, we collected a surrogate soil sample (surface ~3 cm) from a broad-leaf forest at Chubu University campus in central Japan on October 31, 2018. The soil sample was analyzed for LMW monoacids after water extraction employing the analytical protocol described in the experimental section. LMW normal ($C_1$–$C_{10}$), branched ($iC_4$) and hydroxyl monoacids were detected in the soil sample (Kunwar et al., unpublished data, 2018). We found high abundances of formic (7400 ng $g_{wet\ soil}^{-1}$) and acetic (4260 ng

$g_{wet\,soil}^{-1}$) acids in the soil sample, which were significantly higher than the rest of monoacids (~1800 ng $g_{wet\,soil}^{-1}$). Interestingly, hydroxyacids such as glycolic (1680 ng $g_{wet\,soil}^{-1}$) and lactic (1860 ng $g_{wet\,soil}^{-1}$) acids were abundantly detected in the soil samples together with isobutyric acid (77 ng $g_{wet\,soil}^{-1}$) (Kunwar et al., unpublished data, 2018). These preliminary results suggest that monoacids in the forest atmosphere are in part derived from forest soil via microbial decomposition of plant debris and subsequent emission to the air.

However, it is not easy to evaluate the quantitative contribution of monoacids from the forest floor. It is likely that molecular composition of LMW monoacids in soil may depend on a variety of parameters including types of microorganisms in soil, soil organic matter and exudation from plant roots. On the other hand, we consider that photo-oxidation of biogenic VOCs such as isoprene and monoterpenes is an important source of formic and acetic acids in the atmosphere (Paulot et al., 2011)." Please see page 5, lines 5–19.

"Relatively high abundances of particulate lactic and isopentanoic acids were observed in the forest atmosphere (Table 1). A positive correlation was observed between lactic acid and isopentanoic acid in particle phase ($r^2 = 0.98$). Particulate lactic acid did not show correlations with other LMW monoacids detected in particle phase ($r^2 < 0.17$). Isopentanoic acid can be produced by bacteria such as *Clostridium* spp. and *Bacteroides* spp. (Effmert et al., 2012 and references therein). We confirmed that lactic acid is abundantly present in the forest soil from central Japan (1860 ng $g_{wet\,soil}^{-1}$), but isopentanoic acid is below the detection limit (Kunwar et al., unpublished data, 2018). Lactic acid is produced not only by bacteria (*lactobacillus*) (Cabredo et al., 2009) but also by the oxidation of isoprene with ozone (Nguyen et al., 2010). Microflora community in soil system may be different between the two sites; soil-sampling site in central Japan and air-sampling site in northern Japan. More in-depth studies are needed to better understand the emissions of normal, branched and hydroxyl monoacids from forest soil to the atmosphere and interaction between soil and the overlying atmosphere." Please see page 5, lines 27–36.

*4. $C_3$-$C_6$ monoacids will also survive at least for a couple of days (against OH oxidation) and their mention "short-lived" (in section 4) is not supported. For which conclusion do the authors need these sentences (here and also those in page 4, lines 32-35)?*

**Response:**
We deleted a phrase of "short-lived" and the following sentences in revised manuscript.
"Concentrations of long-lived monoacids (formic and acetic acids) in gas phase showed positive correlation with short-lived monoacids ($C_3$-$C_6$)."

"The lifetimes of formic and acetic acids in gas phase are estimated to be 25 and 10 days, respectively (Paulot et al., 2011). These acids can be long range transported in the atmosphere. In gas phase, formic and acetic acids showed positive correlations with short-lived monoacids ($C_3$-$C_6$ monoacids) (day: $r^2 = 0.17$-$0.89$, night: $r^2 = 0.14$-$0.65$)."

We calculated the lifetimes of gaseous $C_1$-$C_4$ and $iC_4$ monoacids with OH radicals (OH radical concentration = $2.0 \times 10^6$ molecule $cm^{-3}$) using the rate constants of gaseous $C_1$-$C_4$ and $iC_4$ monoacids (provided by NIST Chemical Kinetics Database). The lifetimes of these monoacids largely depend on the chain length, but these monoacids are relatively stable (> 2.8 days). Following the comment, the sentences were modified as below.
"To better understand molecular distributions of monoacids in gas phase (i.e., predominance of formic acid followed by acetic acid), we calculated the lifetimes of gaseous $C_1$-$C_4$ and $iC_4$ monoacids with OH radicals (OH radical concentration = $2.0 \times 10^6$ molecule $cm^{-3}$) using the rate constants of gaseous $C_1$-$C_4$ and $iC_4$ monoacids (provided by NIST Chemical Kinetics Database). The lifetimes of gaseous formic, acetic, propionic, butyric and isobutyric acids with OH radicals are 12.9, 8.6, 4.8, 3.2 and 2.8 days.

These results showed that organic acids are relatively stable with longer lifetime for shorter-chain monoacids. This unique feature of lifetime can explain the predominance of formic acid due to the accumulation in gas phase and high concentrations of formic and acetic acids in the atmosphere." Please see page 4, lines 29–35.

*5. Figure 7. Caption is wrong. No data on formic acid are shown.*
**Response:**
Corrected as below.
"Figure 7. Concentrations of acetic acid in particle phase as a function of those of nonanoic acid. The coefficient of determination shows that regression line is statistically significant ($p < 0.05$)."

*I would suggest that the authors need to "clean up" their logic. After revision on individual point raised by the reviewers, clarity of discussion is lacking.*

**Authors' Response to Co-Editor and reviewers**

We appreciate the helpful comments made by Co-Editor and reviewers.

Below, we indicate our point-to-point responses in blue.

**Co-Editor:**

*Comments to the Author:*

*The two reviewer judged the revised manuscript. At least one reviewer still showed great concern about the scientific quality. Particularly the reviewer requests some more concrete evidence that lactic acid is produced by microorganisms, and to argue the possibility of other sources (e.g., photochemistry) more carefully. I find that revision is on the right track but would need to consider if the authors could address to these points.*

*My additional concern is on Figure 9. How to interpret the negative correlation with RH during nighttime? Better not to show regression lines when the correlation is not statistically significant? (also for other figures)*

**Response:**

Based on the reviewer 2' comment, we modified the corresponding sentences in the revised manuscript after the additional laboratory work on the analysis of forest soil samples for monoacids, as follows.

"Relatively high abundances of particulate lactic and isopentanoic acids were observed in the forest atmosphere (Table 1). A positive correlation was observed between lactic acid and isopentanoic acid in particle phase ($r^2 = 0.98$). Particulate lactic acid did not show correlations with other LMW monoacids detected in particle phase ($r^2 < 0.17$). Isopentanoic acid can be produced by bacteria such as *Clostridium* spp. and *Bacteroides* spp. (Effmert et al., 2012 and references therein). We confirmed that lactic acid is abundantly present in the forest soil from central Japan (1860 ng $g_{wet\ soil}^{-1}$), but isopentanoic acid is below the detection limit (Kunwar et al., unpublished data, 2018). Lactic acid is produced not only by bacteria (*lactobacillus*) (Cabredo et al., 2009) but also by the oxidation of isoprene with ozone (Nguyen et al., 2010). Microflora community in soil system may be different between the two sites; soil-sampling site in central Japan and air-sampling site in northern Japan. More in-depth studies are needed to better understand the emissions of normal, branched and hydroxyl monoacids from forest soil to the atmosphere and interaction between soil and the overlying atmosphere." Please see page 5, lines 27–36.

Based on the suggestion, regression line was deleted when the correlation is not statistically significant (please see Figures 6, 7, 8, and 9). We have added the following sentence in figure captions (Figures 6, 7, 8, and 9).

"The coefficient of determination shows that regression line is statistically significant ($p < 0.05$)." Please see Figures 6, 7, and 8.

"The coefficient of determination shows that regression line is statistically significant ($p < 0.1$)." Please see Figure 9.

**Reviewer 1**

*Authors gave sufficient answers to reviewer comments and improved their manuscript based on them. However, I still have some minor concerns listed below:*

*Comment 1. I would think that SO$_4$ is not good tracer for urban air, since general traffic is not usually emitting it. SO$_4$ is mainly emitted by ships and certain industry. Please, correct that. Could you use for example NO$_x$ as an urban traffic tracer?*

**Response:**

NO$_3^-$ is good tracer for urban air. We have modified the sentences in the revised manuscript as follows. "Concentrations of LMW monoacids did not show correlations with anthropogenic tracers such as nss-SO$_4^{2-}$ and NO$_3^-$, indicating that anthropogenic contribution is not important." Please see page 1, lines 17–18.

In addition, we have added the following sentences in the revised manuscript.

"As discussed in the next section, lifetimes of monoacids are relatively long (e.g., 12.9 days for formic acid), suggesting a long-range atmospheric transport of monoacids from other areas. The dominant wind direction was from east and south throughout the sampling period. We compared the concentrations of individual monoacids together with nss-SO$_4^{2-}$ and NO$_3^-$: anthropogenic tracers to evaluate the influence of anthropogenic air mass transport from urban area. We confirmed that individual monoacids in both gas and particle phases did not show any significant correlations with nss-SO$_4^{2-}$ (r$^2$ < 0.14) and NO$_3^-$ (r$^2$ < 0.11). The majority of sampled air was not influenced by urban air masses." Please see page 4, lines 17–23.

*Comment 2. Page 4, line 13: Do you define somewhere, what is short-chain acids and what is long-chain?*

**Response:**

Following the comment, the sentence was modified as below.

"Formic (C$_1$) and acetic (C$_2$) acids are largely present in gas phase." Please see page 4, line 9.

*Comment 3. Check the values in Table 1. At least night time values and F$_p$ for Lactic acid cannot be correct.*

**Response:**

Individual values of $F_p$ were first calculated by each measurement data and then average values of $F_p$ were calculated by individual values of $F_p$. Values of $F_p$ (Table 1) are different from $F_p$ calculated by average concentrations of gaseous and particulate monoacids.

*Comment 4. Page 5, lines 6 and 7: You claim that 'We suggest that gaseous C$_1$–C$_6$ monoacids are emitted from the forest floor where soil microorganisms and plant roots contribute to the emissions of gaseous C$_1$–C$_6$ monoacids'. Based on you data you cannot make this strong conclusion. They can also*

*be produced for example from the reactions of BVOCs.*

**Response:**

Following the comment, we modified the corresponding sentences in the revised manuscript as below.

"Although we did not collect a forest soil sample from Sapporo during the air-sampling period, we collected a surrogate soil sample (surface ~3 cm) from a broad-leaf forest at Chubu University campus in central Japan on October 31, 2018. The soil sample was analyzed for LMW monoacids after water extraction employing the analytical protocol described in the experimental section. LMW normal ($C_1$–$C_{10}$), branched ($iC_4$) and hydroxyl monoacids were detected in the soil sample (Kunwar et al., unpublished data, 2018). We found high abundances of formic (7400 ng $g_{wet\ soil}^{-1}$) and acetic (4260 ng $g_{wet\ soil}^{-1}$) acids in the soil sample, which were significantly higher than the rest of monoacids (~1800 ng $g_{wet\ soil}^{-1}$). Interestingly, hydroxyacids such as glycolic (1680 ng $g_{wet\ soil}^{-1}$) and lactic (1860 ng $g_{wet\ soil}^{-1}$) acids were abundantly detected in the soil samples together with isobutyric acid (77 ng $g_{wet\ soil}^{-1}$) (Kunwar et al., unpublished data, 2018). These preliminary results suggest that monoacids in the forest atmosphere are in part derived from forest soil via microbial decomposition of plant debris and subsequent emission to the air.

However, it is not easy to evaluate the quantitative contribution of monoacids from the forest floor. It is likely that molecular composition of LMW monoacids in soil may depend on a variety of parameters including types of microorganisms in soil, soil organic matter and exudation from plant roots. On the other hand, we consider that photo-oxidation of biogenic VOCs such as isoprene and monoterpenes is an important source of formic and acetic acids in the atmosphere (Paulot et al., 2011)." Please see page 5, lines 5–19.

We have added the following sentence in abstract.

"The forest soil may be a source of gaseous $C_1$–$C_6$ monoacids in the forest atmosphere." Please see page 1, line 20.

In addition, the sentence was modified as below.

"Correlations of $C_1$-$C_6$ monoacids with $iC_4$ suggest that forest floor is a source of gaseous $C_1$–$C_6$ monoacids in the forest atmosphere." Please see page 4, line 42 – page 5, line 1.

*Comment 5. Page 5, line: You claim that 'We suggest that major portion of $C_1$-$C_6$ acids were emitted within the forest floor'. How do you know that it is not from isoprene oxidation or oxidation of some other VOCs? Do not make this strong conclusion.*

**Response:**

Based on comment 4, we have modified the sentences in the revised manuscript. Please see our response to comment 4.

**Reviewer 2**

*General comments:*

*I do understand that the authors cannot provide evidence for the emission of organic acids from the forest soil in this study. Under such circumstances, I expect the authors to provide strong convincing supporting data (for example, soil analysis data, laboratory bacterial experiment etc.) to strengthen the authors' claim. This is especially important when the authors propose a new source of marker compounds. In this revised manuscript the authors failed to do so, and they only provided a small number of biology related references to suggest microorganisms as a source of these acids in the atmosphere. Based on the comments during the discussion phase, and this comments that are substantiated below, I am not able to recommend this manuscript for publication in ACP.*

*Comment 1: Abstract*

*(Revised) Page 1 line 18: Is lactic acid really produced by microorganisms? Do the authors have concrete evidence for this? Correlation does not imply causation.*

**Response:**

Following the comment, we deleted the following sentence in abstract and conclusion.

"Isopentanoic acid in particle phase showed a positive correlation with lactic acid ($r^2 = 0.98$), which is produced by soil microbes."

In addition, we modified the corresponding sentences in the revised manuscript after the additional laboratory work on the analysis of forest soil samples for monoacids. Please see our response to comment 5.

*Discussion*

*Comment 2: (Revised) Page 4 lines 33: This sentence does not make sense. Do the authors mean that OH oxidation is the sole reason for their different concentrations in the atmosphere?*

**Response:**

We modified the corresponding sentences in the revised manuscript. Please see our responses to comments 3 and 5.

*Comment 3: (Revised) Page 4 line 30: These OH rate constants aren't typically accepted values for these compounds. Please refer to NIST Kinetic Database for the list of more widely accepted gas phase OH rate constants of these compounds.*

**Response:**

We calculated the lifetimes of gaseous $C_1$-$C_4$ and $iC_4$ monoacids with OH radicals (OH radical concentration = $2.0 \times 10^6$ molecule cm$^{-3}$) using the rate constants of gaseous $C_1$-$C_4$ and $iC_4$ monoacids (provided by NIST Chemical Kinetics Database). The lifetimes of these monoacids largely depend on the chain length, but these monoacids are relatively stable (> 2.8 days). Following the comment, the sentences were modified as below.

"To better understand molecular distributions of monoacids in gas phase (i.e., predominance of formic acid followed by acetic acid), we calculated the lifetimes of gaseous $C_1$-$C_4$ and $iC_4$ monoacids with OH radicals (OH radical concentration = $2.0 \times 10^6$ molecule cm$^{-3}$) using the rate constants of gaseous $C_1$-$C_4$ and $iC_4$ monoacids (provided by NIST Chemical Kinetics Database). The lifetimes of gaseous formic, acetic, propionic, butyric and isobutyric acids with OH radicals are 12.9, 8.6, 4.8, 3.2 and 2.8 days. These results showed that organic acids are relatively stable with longer lifetime for shorter-chain monoacids. This unique feature of lifetime can explain the predominance of formic acid due to the accumulation in gas phase and high concentrations of formic and acetic acids in the atmosphere." Please see page 4, lines 29–35.

*Comment 4: (Revised) Page 4 line 39: What do you mean by short lived mono acid? What are the lifetimes of C3-C6 acids? From the gas phase OH rate constant of butyric acid ($1.79E^{-12}$ cm$^3$/molecule s), larger acids do not seem to be much short-lived when OH oxidation is solely considered.*

**Response:**

Based on comment 3, we deleted a phrase of "short-lived" and the following sentences in revised manuscript.

"Concentrations of long-lived monoacids (formic and acetic acids) in gas phase showed positive correlation with short-lived monoacids ($C_3$-$C_6$)."

"The lifetimes of formic and acetic acids in gas phase are estimated to be 25 and 10 days, respectively (Paulot et al., 2011). These acids can be long range transported in the atmosphere. In gas phase, formic and acetic acids showed positive correlations with short-lived monoacids ($C_3$-$C_6$ monoacids) (day: $r^2 = 0.17$-$0.89$, night: $r^2 = 0.14$-$0.65$)."

*Comment 5: (Revised) Page 5 lines 1-7: A small number of references to related microbial activities and plant physiology aren't enough to claim a source of marker compounds in atmospheric samples, especially when environmental variables and biological activities can play significant roles in the emissions and transformations of precursor compounds. The authors may not have not taken soil samples during the campaign, but they can always go back later to get some samples, try to understand soil bacteria, or analyze extractable soil organics.*

**Response:**

As suggested, we measured LMW monoacids in soil sample (surface ~3 cm) collected at a broad-leaf forest from Chubu University in Central Japan on October 31, 2018. LMW normal ($C_1$–$C_4$, $C_7$–$C_{10}$), branched ($iC_4$) and hydroxyl (glycolic and lactic) monoacids were detected in the soil sample. We modified the corresponding sentences and added the sentences in the revised manuscript.

"Although we did not collect a forest soil sample from Sapporo during the air-sampling period, we collected a surrogate soil sample (surface ~3 cm) from a broad-leaf forest at Chubu University campus in central Japan on October 31, 2018. The soil sample was analyzed for LMW monoacids after water extraction employing the analytical protocol described in the experimental section. LMW normal ($C_1$–

$C_{10}$), branched ($iC_4$) and hydroxyl monoacids were detected in the soil sample (Kunwar et al., unpublished data, 2018). We found high abundances of formic (7400 ng $g_{wet\ soil}^{-1}$) and acetic (4260 ng $g_{wet\ soil}^{-1}$) acids in the soil sample, which were significantly higher than the rest of monoacids (~1800 ng $g_{wet\ soil}^{-1}$). Interestingly, hydroxyacids such as glycolic (1680 ng $g_{wet\ soil}^{-1}$) and lactic (1860 ng $g_{wet\ soil}^{-1}$) acids were abundantly detected in the soil samples together with isobutyric acid (77 ng $g_{wet\ soil}^{-1}$) (Kunwar et al., unpublished data, 2018). These preliminary results suggest that monoacids in the forest atmosphere are in part derived from forest soil via microbial decomposition of plant debris and subsequent emission to the air.

However, it is not easy to calculate the quantitative contribution of monoacids from the forest floor. It is likely that molecular composition of LMW monoacids in soil may depend on a variety of parameters including types of microorganisms in soil, soil organic matter and exudation from plant roots. On the other hand, we consider that photo-oxidation of biogenic VOCs such as isoprene and monoterpenes is an important source of formic and acetic acids in the atmosphere (Paulot et al., 2011)." Please see page 5, lines 5–19.

"Relatively high abundances of particulate lactic and isopentanoic acids were observed in the forest atmosphere (Table 1). A positive correlation was observed between lactic acid and isopentanoic acid in particle phase ($r^2 = 0.98$). Particulate lactic acid did not show correlations with other LMW monoacids detected in particle phase ($r^2 < 0.17$). Isopentanoic acid can be produced by bacteria such as *Clostridium* spp. and *Bacteroides* spp. (Effmert et al., 2012 and references therein). We confirmed that lactic acid is abundantly present in the forest soil from central Japan (1860 ng $g_{wet\ soil}^{-1}$), but isopentanoic acid is below the detection limit (Kunwar et al., unpublished data, 2018). Lactic acid is produced not only by bacteria (*lactobacillus*) (Cabredo et al., 2009) but also by the oxidation of isoprene with ozone (Nguyen et al., 2010). Microflora community in soil system may be different between the two sites; soil-sampling site in central Japan and air-sampling site in northern Japan. More in-depth studies are needed to better understand the emissions of normal, branched and hydroxyl monoacids from forest soil to the atmosphere and interaction between soil and the overlying atmosphere." Please see page 5, lines 27–36.

*Comment 6: (Revised) Page 5 lines 24: "somewhat end products" does not makes sense. It is either "end product" or "intermediate".*

**Response:**

We deleted the following sentence in the revised manuscript.

"Formic and acetic acids are intermediate products in the oxidative degradation of various VOCs. They may be very stable intermediate products before the oxidation to $CO_2$."

We have checked references. We made a mistake in the author name.

We have replaced "Uta et al., 2012" by "Effmert et al., 2012".

We have added the following papers in the reference section.

Kawamura, K. and Gagosian, R. B.: Implication of ω-oxocarboxylic acids in the remote marine atmosphere for photooxidation of unsaturated fatty acids, Nature, 325, 330-332, 1987.

Tsai, Y. I. and Kuo, S. C.: Contributions of low molecular weight carboxylic acids to aerosols and wet deposition in a natural subtropical broad-leaved forest environment, Atmos. Environ., 81, 270-279, 2013.

---

## Author Response (AR4)

**Authors' Response to Co-Editor**

We appreciate the helpful comment made by Co-Editor. Below, we indicate our point-to-point response in blue.

*Previous concerns were mostly adequately addressed. But one important point is not fully resolved, as to if the soil analysis supports the conclusion, which the two reviewers thought critical. I still recommend performing ON-SITE soil sampling in the similar season. Otherwise, the authors should provide strong justification of similarity in the environment of the two places.*

**Response:**

Based on the comment, we added the following sentences in the revised manuscript.

"The forest floor at Chubu University site is similar to that of Sapporo site in terms of the coverage with a broad-leaf litter from similar plant species including a Japanese oak. The climate in central Japan is different from northern Japan, but the air temperature recorded in Nagoya next to the Chubu University campus in October 2018 (average: 19±2.9 °C, Japan Meteorological Agency: https://www.jma.go.jp/jma/indexe.html) was similar to that of Sapporo (21±2.3 °C) during the air-sampling period. These similarities provide a strong justification to utilize the soil sample from Chubu University site as a surrogate of Sapporo forest soil." Please see page 5, lines 8-14.